# A network of cytosolic (co)chaperones promotes the biogenesis of mitochondrial signal-anchored outer membrane proteins

Layla Drwesh[1], Benjamin Heim[2], Max Graf[1], Linda Kehr[1], Lea Hansen-Palmus[1], Mirita Franz-Wachtel[3], Boris Macek[3], Hubert Kalbacher[1], Johannes Buchner[2], Doron Rapaport[1]*

[1]Interfaculty Institute of Biochemistry, University of Tübingen, Tuebingen, Germany; [2]Center for Integrated Protein Science, Department of Chemistry, Technische Universität München, Garching, Germany; [3]Proteome Center Tübingen, Interfaculty Institute for Cell Biology, University of Tübingen, Tübingen, Germany

**Abstract** Signal-anchored (SA) proteins are anchored into the mitochondrial outer membrane (OM) via a single transmembrane segment at their N-terminus while the bulk of the proteins is facing the cytosol. These proteins are encoded by nuclear DNA, translated on cytosolic ribosomes, and are then targeted to the organelle and inserted into its OM by import factors. Recently, research on the insertion mechanisms of these proteins into the mitochondrial OM have gained a lot of attention. In contrast, the early cytosolic steps of their biogenesis are unresolved. Using various proteins from this category and a broad set of in vivo, *in organello*, and in vitro assays, we reconstituted the early steps of their biogenesis. We identified a subset of molecular (co)chaperones that interact with newly synthesized SA proteins, namely, Hsp70 and Hsp90 chaperones and co-chaperones from the Hsp40 family like Ydj1 and Sis1. These interactions were mediated by the hydrophobic transmembrane segments of the SA proteins. We further demonstrate that interfering with these interactions inhibits the biogenesis of SA proteins to a various extent. Finally, we could demonstrate direct interaction of peptides corresponding to the transmembrane segments of SA proteins with the (co)chaperones and reconstitute in vitro the transfer of such peptides from the Hsp70 chaperone to the mitochondrial Tom70 receptor. Collectively, this study unravels an array of cytosolic chaperones and mitochondrial import factors that facilitates the targeting and membrane integration of mitochondrial SA proteins.

**\*For correspondence:**
doron.rapaport@uni-tuebingen.de

**Competing interest:** The authors declare that no competing interests exist.

## Editor's evaluation

The authors dissect and reconstitute the cytosolic steps of biogenesis of mitochondrial signal-anchored membrane proteins, focusing on post-translational precursor recognition by cytosolic chaperones and their subsequent transfer to mitochondrial import receptors. These are crucial events in order to assist proper protein biogenesis while preventing aggregation and its downstream consequences. The study is an important contribution to the understanding of cytosolic events in the biogenesis of mitochondrial proteins and will be of relevance for researchers in the fields of chaperone and mitochondrial biology as well as for cell biologists studying the biogenesis of membrane proteins.

## Introduction

Even though mitochondria have their own genome, the vast majority of their proteins are encoded by the nuclear genome, synthesized on cytosolic ribosomes, and then imported into the organelle. The early stages of these pathways are believed to be mediated by cytosolic factors and chaperones, whereas the later ones are facilitated by protein import machineries that have evolved in the different mitochondrial compartments (*Neupert and Herrmann, 2007*; *Wiedemann and Pfanner, 2017*).

The mitochondrial outer membrane harbors proteins with variable topologies that can span the membrane once, twice, or as multi-span proteins. Proteins that span the membrane once can have their single transmembrane segment (TMS) in the center of the protein or at the N- or C- terminus (*Drwesh and Rapaport, 2020*; *Gupta and Becker, 2021*). The latter group is called the tail-anchored proteins whereas those anchored via an N-terminal segment are known as signal-anchored (SA) proteins. Some known SA proteins are the TOM receptors Tom70 and Tom20, the quality control protein Msp1 (*Okreglak and Walter, 2014*; *Chen et al., 2014*), and the mitochondrial outer membrane isoform of Mcr1 (Mcr1$_{mom}$) (*Lamb et al., 1999*).

Despite sharing similar protein topology, SA proteins do not appear to follow the same insertion route. While insertion of Tom70 and Tom20 was previously reported to be dependent on the MIM complex and elements of the TOM complex (*Ahting et al., 2005*; *Becker et al., 2008*; *Popov-Celeketić et al., 2008*), Msp1 mitochondrial insertion was shown to require only the MIM complex. The biogenesis of another member of the group Mcr1$_{mom}$ has been proposed to be independent of the TOM complex, although it is yet unclear whether the MIM complex is involved (*Meineke et al., 2008*; *Vitali et al., 2020*; *Doan et al., 2020*).

Like the rest of the mitochondrial OM proteins, SA proteins, are initially synthesized in the cytosol before being targeted to mitochondria. Such proteins contain an exposed hydrophobic transmembrane segment, which makes them vulnerable to aberrant folding and aggregation. This situation can potentially result in cytotoxic protein species, which might contribute to the pathomechanism of various neurodegenerative and other diseases (*Bohush et al., 2019*: *Chaari, 2019*). Hence, it is widely thought that cytosolic factors bind such newly synthesized proteins, thereby maintaining their import-competent conformation by counteracting aggregation, degradation, and misfolding (*Neupert and Pfanner, 1993*).

In the yeast *Saccharomyces cerevisiae*, a large repertoire of molecular chaperones was identified that regulate protein quality control. These elements are classified into different families according to their molecular masses and the way they interact with their substrate (*Mokry et al., 2015*). Chaperones from the Hsp100 family have high-binding affinity to aggregated proteins and function as disaggregases by both reactivating and resolubilizing them (*Zolkiewski et al., 2012*). Chaperones from the Hsp70 family (like Ssa1-4 in yeast) function in a wide range of biological processes, such as modulating folding and preventing aggregation. Chaperones from this family associate with a broad spectrum of client proteins in an ATP-regulated cycle. Client protein recognition is regulated mainly by J-proteins (co-chaperones like Ydj1 and Sis1 in yeast) from the Hsp40 family that stimulate ATP hydrolysis, thereby facilitating client capture by Hsp70 (*Cyr, 1995*; *Kampinga and Craig, 2010*; *Wyszkowski et al., 2021*). Beside their crucial role in modulating the ATPase cycle of Hsp70 through their J-domain, Hsp40 chaperones can also bind unfolded protein substrates (*Johnson and Craig, 2001*; *Li et al., 2009*). In addition, cells harbor small heat shock proteins (sHSPs) that bind non-native proteins and are crucial for preventing irreversible aggregation processes. Such sHSPs were recently found to be involved also in protecting proteins from mechanical stress (*Haslbeck et al., 2019*; *Collier and Benesch, 2020*). In yeast, Hsp26 and Hsp42 are chaperones from this family that have been reported to associate with cytosolic aggregates allowing the Ssa1-Hsp104 chaperone system to efficiently disassemble and refold them (*Haslbeck et al., 2005*; *Haslbeck et al., 2004*; *Cashikar et al., 2005*).

Chaperones of the Hsp70 family as well as some of their co-chaperones were implicated in the import of mitochondrial presequence-containing substrates (*Hoseini et al., 2016*; *Xie et al., 2017*; *Endo et al., 1996*; *Deshaies et al., 1988*; *Caplan et al., 1992*) and carrier proteins of the inner membrane (*Young et al., 2003*; *Bhangoo et al., 2007*). For example, the co-chaperone Djp1 plays a key role in the ER-SURF pathway which involves a de-tour of mitochondrial substrates to the ER (*Hansen et al., 2018*). Such (co)chaperones were also reported to facilitate the import of mitochondrial OM proteins like Mim1 and Tom22 (*Papić et al., 2013*; *Opaliński et al., 2018*). Furthermore, we previously reported that cytosolic Hsp70 and Hsp40 chaperones enable the biogenesis of

mitochondrial β-barrel proteins (*Jores et al., 2018*). Pex19, which is a cytosolic chaperone associated with the peroxisomal import of membrane proteins, has also been found, along with Ssa1 and its co-chaperone Sti1, to assist the biogenesis of mitochondrial tail-anchored proteins namely, Fis1 and Gem1 (*Cichocki et al., 2018*; *Jansen and van der Klei, 2019*).

Despite this advancement in our understanding of the contribution of cytosolic chaperones to mitochondrial biogenesis, no cytosolic factors have been reported so far as mediators of the biogenesis of mitochondrial SA proteins. Currently, there is scarce information regarding the early cytosolic events that assure their safe passage through the cytosol. To fill this gap, we have employed a combined experimental strategy consisting of assays with yeast cells extract, isolated organelles, and in vitro experiments with purified proteins. We could identify a subset of Hsp70 and Hsp40 (co) chaperones that interacts with SA proteins. These (co)chaperones were further shown to be crucial for the mitochondrial import of SA proteins. Furthermore, we suggest a novel offsetting role of the import receptors Tom20 and Tom70 in promoting the insertion of SA proteins with the latter serving as a docking site for the Hsp70 chaperones.

## Results
### Cytosolic chaperones interact with newly synthesized signal-anchored proteins

Signal-anchored proteins are, due to their hydrophobic segments, at a high risk of aggregation and misfolding following their translation in the cytosol. We aimed to search for factors that can prevent such a scenario and maintain the SA substrates in an import-competent conformation. We chose four SA model proteins to study the potential involvement of such cytosolic components: the two receptor subunits of the TOM complex, Tom20 and Tom70, and two additional proteins namely, Msp1 and the OM isoform of Mcr1 (Mcr1$_{mom}$). Initially, we wanted to determine which factors can interact with newly synthesized SA proteins. To this end, we used yeast extract to translate in vitro signal-anchored proteins with a C-terminally 3HA-tag. Since the yeast extract does not contain organelles to where the freshly translated proteins can be targeted, we anticipated that such SA proteins should associate in the hydrophilic environment of the extract with factors, which will maintain them in an import-competent conformation. The tail-anchored protein Fis1 and the β-barrel protein Porin were included for comparison and the unrelated protein dihydrofolate reductase (DHFR) was used as a control. After their synthesis, all proteins were pulled-down with anti-HA beads and co-purified proteins in the elution fraction were analyzed by western blot followed by immunodecoration with antibodies against different cytosolic elements (*Figure 1A and B*).

All four signal-anchored proteins co-eluted with the Hsp70 chaperones Ssa1/2, and the Hsp40 co-chaperones Ydj1, Sis1 and Djp1. Weaker interactions were observed with the Hsp90 chaperones Hsc82/Hsp82 and their co-chaperones Aha1 and Sti1. The eluates contained also Hsp104 chaperone and marginal amounts of Hsp42 chaperone, suggesting that a minor portion of the newly synthesized proteins got aggregated. The co-chaperone Hch1 and the cytosolic protein Bmh1 were not co-eluted with any of the tested proteins (*Figure 1A and B*). Control elution fraction containing newly synthesized DHFR had neglectable amounts of co-purified (co)chaperones, indicating the binding specificity of chaperone-substrate. As we observed previously (*Jores et al., 2018*), the β-barrel protein Porin also associated with the various (co)chaperones.

To validate the western blot results and to search for additional (co)chaperones co-purified with the newly translated SA proteins, the elution fractions from such pull-down assays were analyzed also by mass spectrometry. The mass spectrometry analysis of the eluates from pull-downs with Msp1, Mcr1, or (as a control) with mock pull-down where no synthesized protein was present are shown in *Supplementary file 1*. Several chaperones, which are not included in the western blot due to lack of antibodies, were additionally detected in the eluate of both proteins Msp1 and Mcr1. The detected proteins are presented in two groups. The first contained those proteins that were not found at all in the mock eluate, hence their enrichment ratio in Msp1 and Mcr1 eluates could not be calculated (*Supplementary file 1A*). Among these, chaperones from the Hsp70 family, namely, Ssa4, Snl1, and Lhs1 were found. The second group is the set of proteins that were found in minor amounts also in the mock elution, and thus their relative enrichment as compared to the mock pull-down could be determined (*Supplementary file 1B*). Amongst them, the Hsp70 chaperones Ssc1, Sse1, and Ssb1/2, in

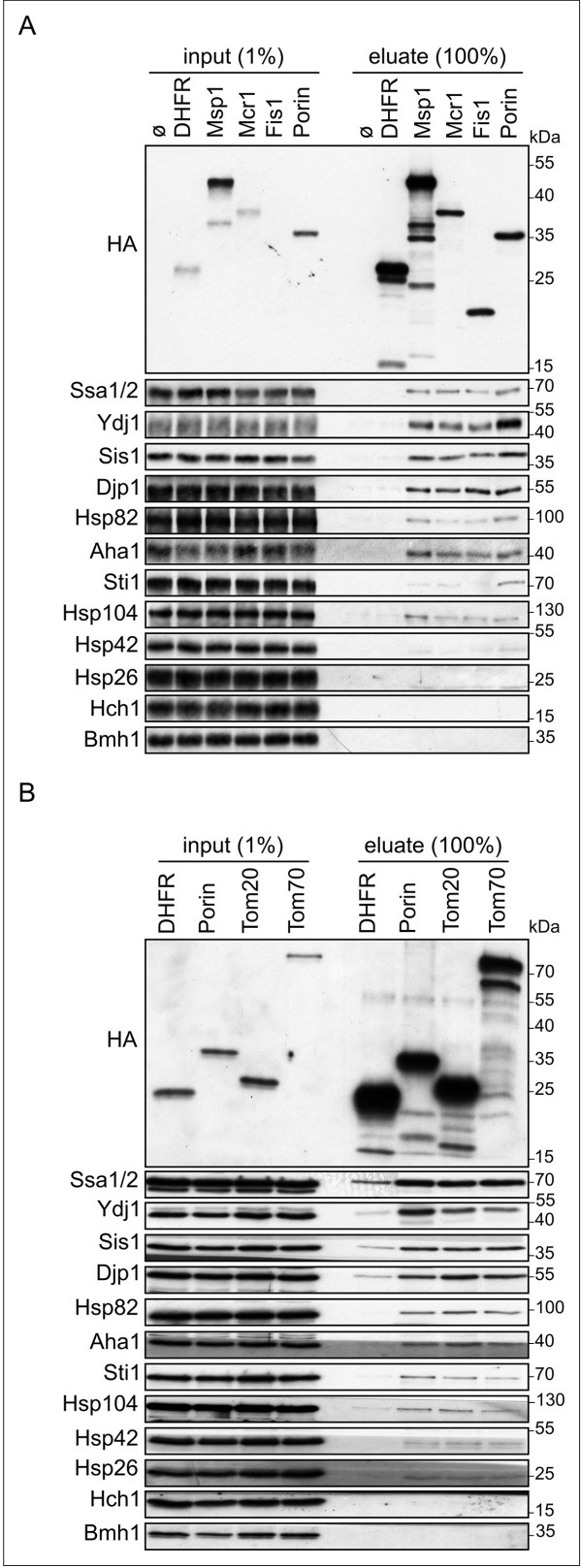

**Figure 1.** Cytosolic chaperones interact with newly synthesized signal-anchored proteins. (**A and B**) In vitro translation reactions included yeast extracts without mRNA (Ø) or programmed with mRNA encoding HA-tagged variants of signal-anchored proteins (Msp1, Mcr1, Tom20, and Tom70), the tail-anchored protein Fis1, the β-barrel protein Porin, or, as a control, dihydrofolate reductase (DHFR). The reactions were subjected to a pull-down

*Figure 1 continued on next page*

*Figure 1 continued*

with anti-HA beads. Samples from the input (1%) and the eluates (100%) were analyzed by SDS-PAGE and immunodecoration with the indicated antibodies.

The online version of this article includes the following source data for figure 1:

**Source data 1.** Source data for *Figure 1A*.

**Source data 2.** Source data for *Figure 1B*.

addition to the ribosome-associated complex (RAC) chaperone Zuo1 were detected. Taken together, the combined analysis of the pull-down assays shows multifaceted interactions between newly synthesized SA proteins and cytosolic chaperones and co-chaperones.

## The hydrophobic transmembrane segment of SA proteins mediates their interactions with cytosolic (co)chaperones

The detected interactions of a subset of cytosolic (co)chaperones with SA proteins led us to ask which part of these latter proteins mediate such association. A likely candidate for this task is the proteins' hydrophobic TMS. To test this assumption, we constructed two additional C-terminally HA-tagged versions of each SA protein – one construct encodes the cytosolic part of the protein (indicated by protein name-C, *Figure 2*) and the second encodes only the hydrophobic TMS (indicated by protein name-T, *Figure 2*). These two constructs together with the full-length (-FL) variant were used for in vitro translation in yeast extract followed by pull-down assay. As expected for small proteins, all three constructs encoding the hydrophobic TMSs were synthesized to a lower extent as compared to the constructs representing the cytosolic moieties or the full-length proteins (*Figure 2*, input panels). Yet, similar levels of bound (co)chaperones were observed in the eluates of the hydrophobic TMSs and the full-length versions (*Figure 2A–C*). In contrast, only marginal binding of (co)chaperones to the soluble cytosolic parts of the SA proteins or to the control protein DHFR was detected (*Figure 2A–C*). Taken together, our findings suggest that the interactions between SA proteins and molecular chaperones are mainly governed by the TMSs of the mitochondrial SA proteins.

## Signal-anchored proteins show variable dependence on Hsp40 co-chaperones

After demonstrating that Hsp40 co-chaperones like Ydj1 and Sis1 can physically interact with newly synthesized SA proteins, we wanted to test if these interactions are relevant to the biogenesis of proteins belonging to the latter group. These two co-chaperones were previously reported to be involved in the biogenesis of the β-barrel protein Porin (*Jores et al., 2018*). Moreover, Ydj1 has been implicated in the mitochondrial import of presequence-containing proteins (*Caplan et al., 1992*; *Xie et al., 2017*). To gain insight into the physiological relevance of these co-chaperones for the biogenesis of signal-anchored proteins, we created strains expressing *YDJ1*, *SIS1* or both genes under the control of a tetracycline-repressible promoter. To accelerate the depletion process, ubiquitin was fused N-terminally to the down-regulated protein. To monitor the effect of depleted chaperones over time, cells were grown for two hours in the absence of doxycycline. Then, down-regulation was induced by supplementing doxycycline (Dox) to the medium, and cells were harvested immediately (time = 0) or after 1, 2, or 4 hr. Cytosolic and mitochondria fractions were later obtained from the harvested cells.

As anticipated, cytosolic levels of Ydj1 and Sis1 were gradually decreased over time after addition of doxycycline, whereas levels of the other chaperones like Hsp104 and Hsp26 remained unchanged, demonstrating doxycycline´s selectivity in suppressing the expression of only the genes regulated by the tetracycline promoter (*Figure 3B and E*). Inspecting the mitochondrial fractions revealed that none of the examined mitochondrial proteins was altered in Ydj1 depleted cells. In contrast, mitochondrial levels of Tom20, Tom70, and the IMS form of Mcr1 were moderately reduced after 4 hr in Sis1 depleted cells , while no change was detected in the levels of the other signal-anchored proteins Msp1 and the OM form of Mcr1, or other outer membrane proteins like Porin or Fis1 (*Figure 3A and C*). These observations might be explained by the previous report that the J-domain of Sis1 can compensate for the loss of Ydj1 J-domain but not the other way around (*Yan and Craig, 1999*).

To avoid such cross compensation of the two co-chaperones, we wanted next to check the effect of the parallel depletion of both on the biogenesis of SA proteins. Interestingly, the levels of Tom20

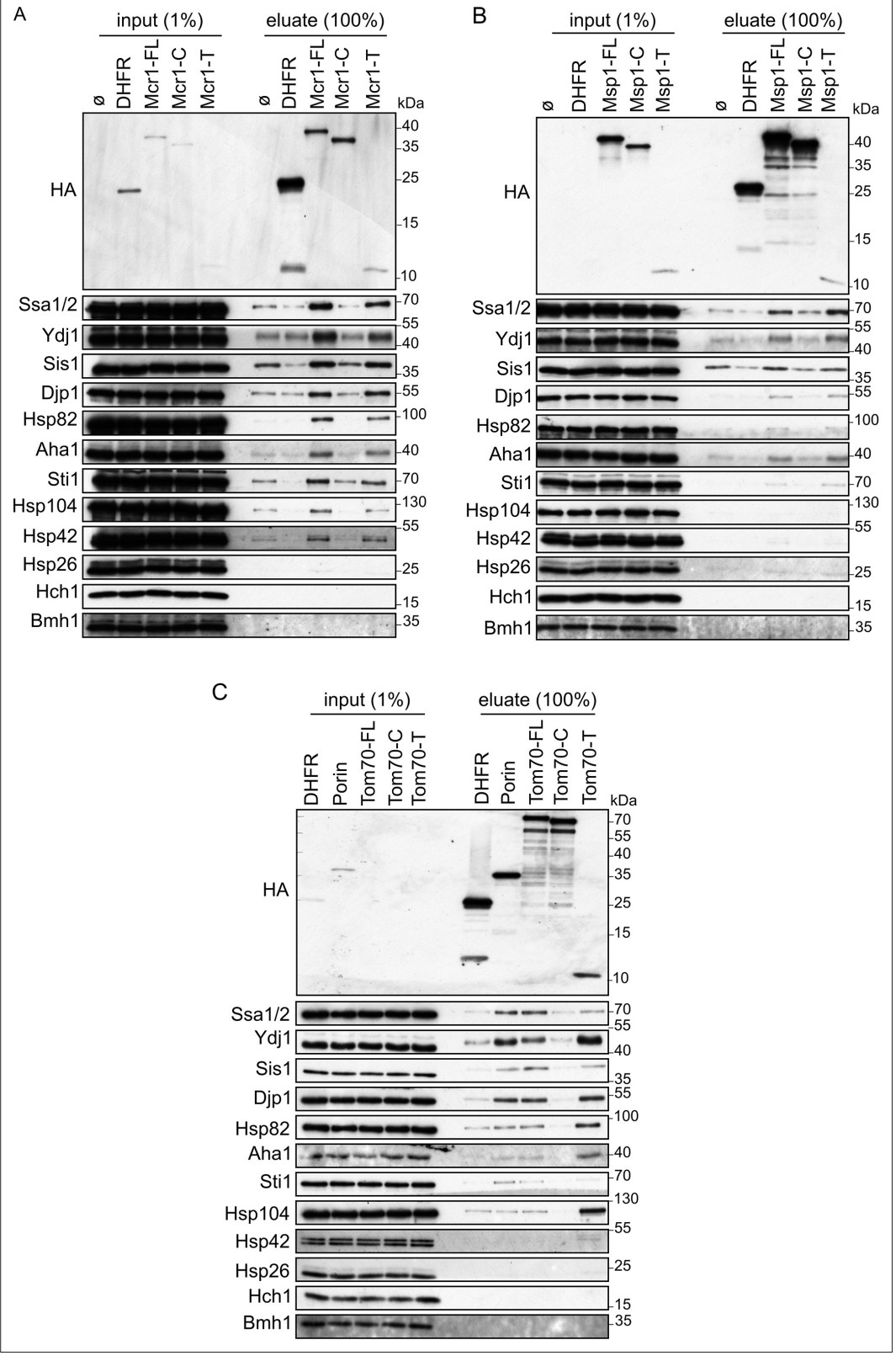

**Figure 2.** Cytosolic chaperones interact with newly synthesized signal-anchored proteins through their transmembrane segment. (**A–C**) In-vitro translation reactions included yeast extracts without mRNA (Ø) or programmed with mRNA encoding HA-tagged versions of DHFR or the full length (FL), cytosolic domain (**C**) or transmembrane segment (**T**) of the SA proteins: Mcr1 (**A**), Msp1 (**B**) and Tom70 (**C**). The reactions were subjected to

*Figure 2 continued on next page*

*Figure 2 continued*

a pull-down with anti-HA beads. Samples from the input (1%) and the eluates (100%) were analyzed by SDS-PAGE and immunodecoration with the indicated antibodies.

The online version of this article includes the following source data for figure 2:

**Source data 1.** Source data for *Figure 2A*.

**Source data 2.** Source data for *Figure 2B*.

**Source data 3.** Source data for *Figure 2C*.

and Tom70 were gradually reduced over time down to 40–50% of their levels in control conditions, the amounts of Mcr1 protein were only slightly affected ,while levels of Msp1 were not altered. As expected from our previous study (*Jores et al., 2018*), Porin levels were also showed a decrease to 40% (*Figure 3D and F*). To exclude any indirect effect that might be caused by a general dysfunction of the mitochondria or by defective assembly of the known import machineries due to low expression levels of Ydj1 and Sis1, we monitored the steady state levels and the assembly of additional mitochondrial proteins from different sub-compartments (*Figure 3D*, *Figure 3—figure supplement 1A*, B). Some of these proteins are subunits of the respiratory chain complexes, such as Cox2 of complex IV and Cor1 of complex III, others are components of the OM import machineries: Mim1, Tom40, and Tom22, or subunit of the IM import machinery like Oxa1.

Interestingly, mitochondrial levels of the α-helical OM proteins Tom22 and Om14 and the IM protein Oxa1 were also reduced in the depletion strain (*Figure 3D*). We cannot exclude the possibility that this reduction, and especially that of Tom22, might be a result of lower levels of the TOM receptors Tom20 and Tom70, since both receptors were shown to play a role in the membranal insertion of Tom22 (*Keil and Pfanner, 1993*). On the other hand, the levels of the respiratory subunits Cox2 and Cor1, the matrix protein Aco1, the IM carrier protein Pic2, and the MIM complex subunit Mim1 remained unaffected (*Figure 3D*, *Figure 3—figure supplement 1A*). Additionally, the assembly, as analyzed by BN-PAGE, of the respiratory chain complexes III and IV and the MIM complex were not altered in the depletion strain whereas that of the assembled TOM core complex was slightly reduced (*Figure 3—figure supplement 1B*). We suggest that this compromised assembly of the TOM complex is due to the known decreased steady state levels of Tom40 upon depletion of Ydj1 and Sis1 (*Figure 3—figure supplement 1A* and *Jores et al., 2018*).

These results indicate that the observed lower levels of some mitochondrial proteins, including those of Tom20 and Tom70, in Ydj1 and Sis1 depleted strain are not caused by a general defect in mitochondrial import or compromised assembly of the MIM complex, which is implicated in the biogenesis of α-helical proteins. Altogether, these results suggest that biogenesis of various SA proteins rely to a different degree on Hsp40 co-chaperones.

The highly reduced steady-state levels of Tom20 and Tom70 upon depletion of both co-chaperones led us to conduct in vitro import assays. In these experiments, we translated radiolabeled variants of Tom20 and Tom70 in yeast extract from either control or Ydj1 and Sis1 depleted cells and incubated the newly synthesized proteins with isolated mitochondria. We found that extract depleted of both co-chaperones can support to a lower extent the import of Tom20 but to a normal degree that of Tom70 (*Figure 4A and B*). Thus, it seems that the import of Tom70 under these in vitro conditions can be compensated by other, yet unknown, factors.

## Depletion of Ydj1 and Sis1 can increase the risk for aggregation of newly synthesized proteins

To better understand the involvement of Ydj1 and Sis1 in the cytosolic maintenance of signal-anchored proteins, we aimed to analyze the interaction pattern between the other chaperones and the newly synthesized SA proteins in yeast cells depleted of Ydj1 and Sis1. We choose to utilize a strain that is depleted of both co-chaperones since their mutual function allows them to compensate for each other's loss. To this end, we translated HA-tagged variants of the four signal-anchored proteins in extract of either WT or Ydj1 and Sis1 depleted cells (YS↓) and then performed pull down with anti-HA beads. Interestingly, we found that the interaction between the newly synthesized proteins and some of the chaperones was altered in cells depleted for both co-chaperones while other chaperones did not show any considerable change. The co-purified levels of Hsp104 and Hsp26 chaperones were

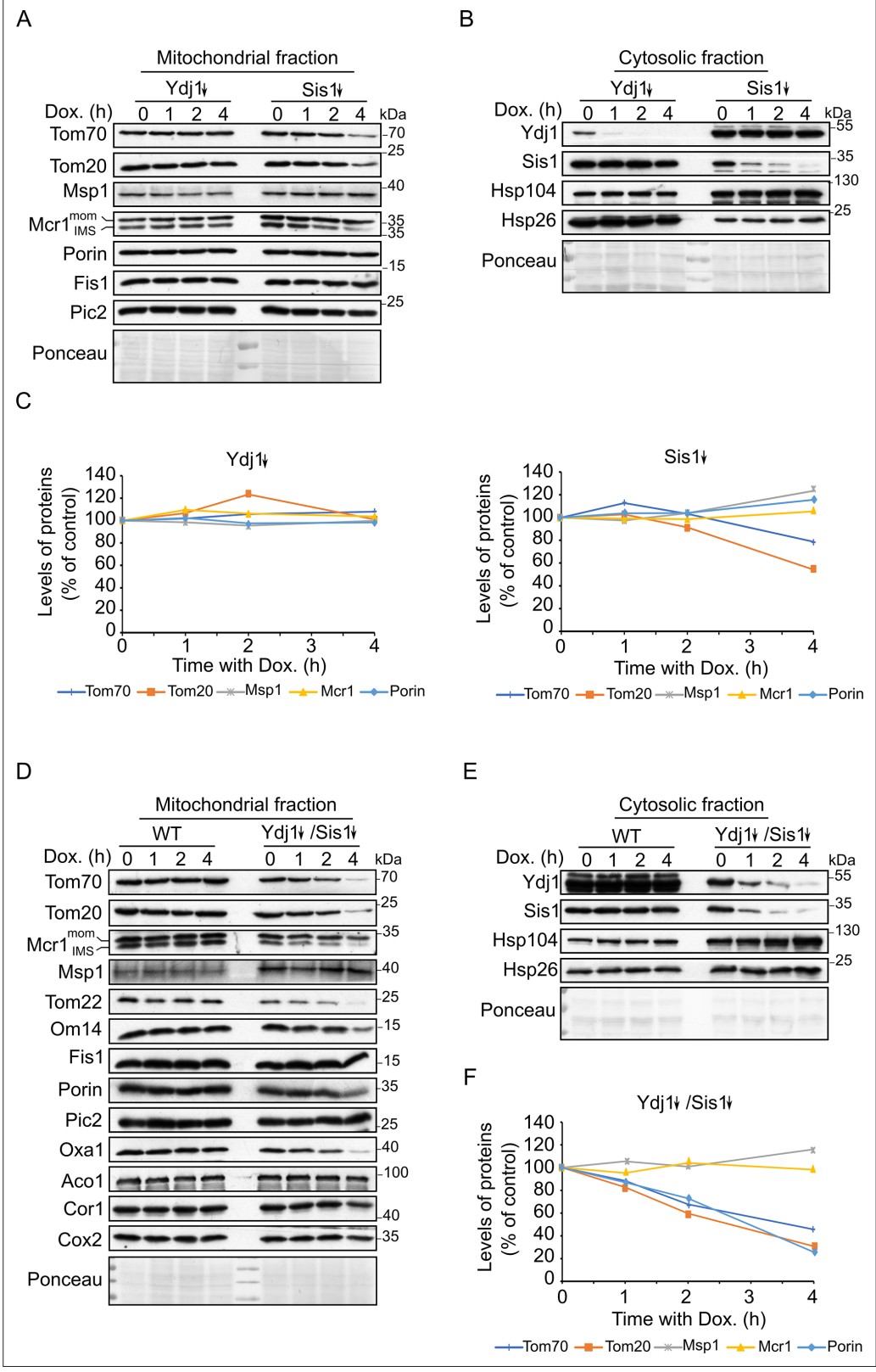

**Figure 3.** Depletion of both co-chaperones Ydj1 and Sis1 results in decreased steady-state levels of Tom20 and Tom70. (**A, B, D and E**) Mitochondrial (**A and D**) and cytosolic (**B and E**) fractions were isolated from WT cells, cells depleted for either Ydj1 (Ydj1↓) or Sis1 (Sis1↓), or from cells double depleted for both co-chaperones (Ydj1↓Sis1↓). Cells were grown without doxycycline (time = 0) or in the presence of Dox for 1, 2, or 4 hr. Samples were analyzed

*Figure 3 continued on next page*

*Figure 3 continued*

by SDS-PAGE followed by immunodecoration with the indicated antibodies. (**C and F**) Intensities of the bands corresponding to the depicted proteins in the mitochondrial fractions from three independent experiments were quantified and normalized to Ponceau levels. The levels of the proteins in each depletion strain in the absence of doxycycline (time = 0) was set to 100%. Error bars represent ± SD.

The online version of this article includes the following source data and figure supplement(s) for figure 3:

**Source data 1.** Source data for *Figure 3A and B*.

**Source data 2.** Source data for *Figure 3D*.

**Source data 3.** Source data for *Figure 3E*.

**Source data 4.** Raw data for plot in *Figure 3C*.

**Source data 5.** Raw data for plot in *Figure 3F*.

**Figure supplement 1.** Depletion of both Ydj1 and Sis1 does not alter the assembly of the respiratory chain complexes or of the import machineries.

**Figure supplement 1—source data 1.** Source data for *Figure 3—figure supplement 1A*.

**Figure supplement 1—source data 2.** Source data for *Figure 3—figure supplement 1B*.

increased in the eluates of all SA proteins translated in the extract of the depletion strain (*Figure 5A and B*). Both Hsp104 and Hsp26 are involved in disaggregation of substrate proteins. Hence the co-purification of these chaperones with newly synthesized SA proteins, in the absence of Sis1 and Ydj1, suggests that the cytosol of such strain offers less stabilization for the newly synthesized substrates, and as a result, these substrates are more prone to aggregation under these conditions. These findings indicate a role of the Hsp40 co-chaperones in keeping the signal-anchored proteins stable in the cytosol.

## The binding affinity of substrate to Hsp70 chaperone Ssa1 is higher than the affinity to the Hsp40 co-chaperone Sis1

To better understand the dynamics of chaperone-substrate interactions, we next aimed to investigate their binding kinetics and affinity using fluorescence anisotropy. Since we have shown that the interaction between chaperones and signal-anchored proteins is mediated by their transmembrane domain (*Figure 2*), we synthesized peptides corresponding to residues 1–39 and 1–32 of Mcr1 and Tom70, respectively, which contain the TMS, and modified the peptides with the fluorescent dye Tetramethylrhodamine (TAMRA). Then, we monitored the anisotropy changes of these fluorescently labeled peptides upon their interaction with recombinantly expressed and purified (co)chaperones. The fluorescence anisotropy of the TAMRA-labeled Tom70 and Mcr1 peptides increased upon addition of recombinant Ssa1 indicating the formation of a peptide-Ssa1 complex. As a control, such an increase was not observed when bovine serum albumin (BSA) was added, underlining the specificity of the chaperone-substrate interaction (*Figure 6A and B*). These observations demonstrate that transmembrane segments of SA proteins can bind directly to Ssa1.

Next, we implemented titration assay in which increasing concentrations of Ssa1 (*Figure 6C and D*) or Sis1 (*Figure 6E and F*) were sequentially added to either Tom70 or Mcr1 peptides. This approach enabled us to monitor the binding parameters and compare the binding affinity between substrate and different (co)chaperones. Based on these experiments, dissociation constants (Kd) were determined according to hyperbolic regression curves fitted to the data. These Kds were calculated to be 2.42 µM for Ssa1-Tom70(TMS) complex and 3.27 µM for Ssa1-Mcr1(TMS) complex (*Figure 6C and D*). Interestingly, dissociation constants (Kd) of 18.3 µM and 25.75 µM were measured for Sis1-Tom70(TMS) complex and Sis1-Mcr1(TMS) complex, respectively (*Figure 6E and F*). Collectively, these results show that the affinity between the substrates and the Hsp70 chaperone Ssa1 is several folds higher than their affinity to the co-chaperone, Sis1. This observation supports the notion that the co-chaperone facilitates the initial low-affinity interaction with the substrate before passing it to the main chaperone to which it has a higher affinity.

## Hsp70 chaperones are required for optimal biogenesis of SA proteins

Hsp70 chaperones cooperate with different cofactors and are known to be regulated mainly by J-protein co-chaperones like Ydj1 and Sis1. Our results so far have shown that these chaperones play an

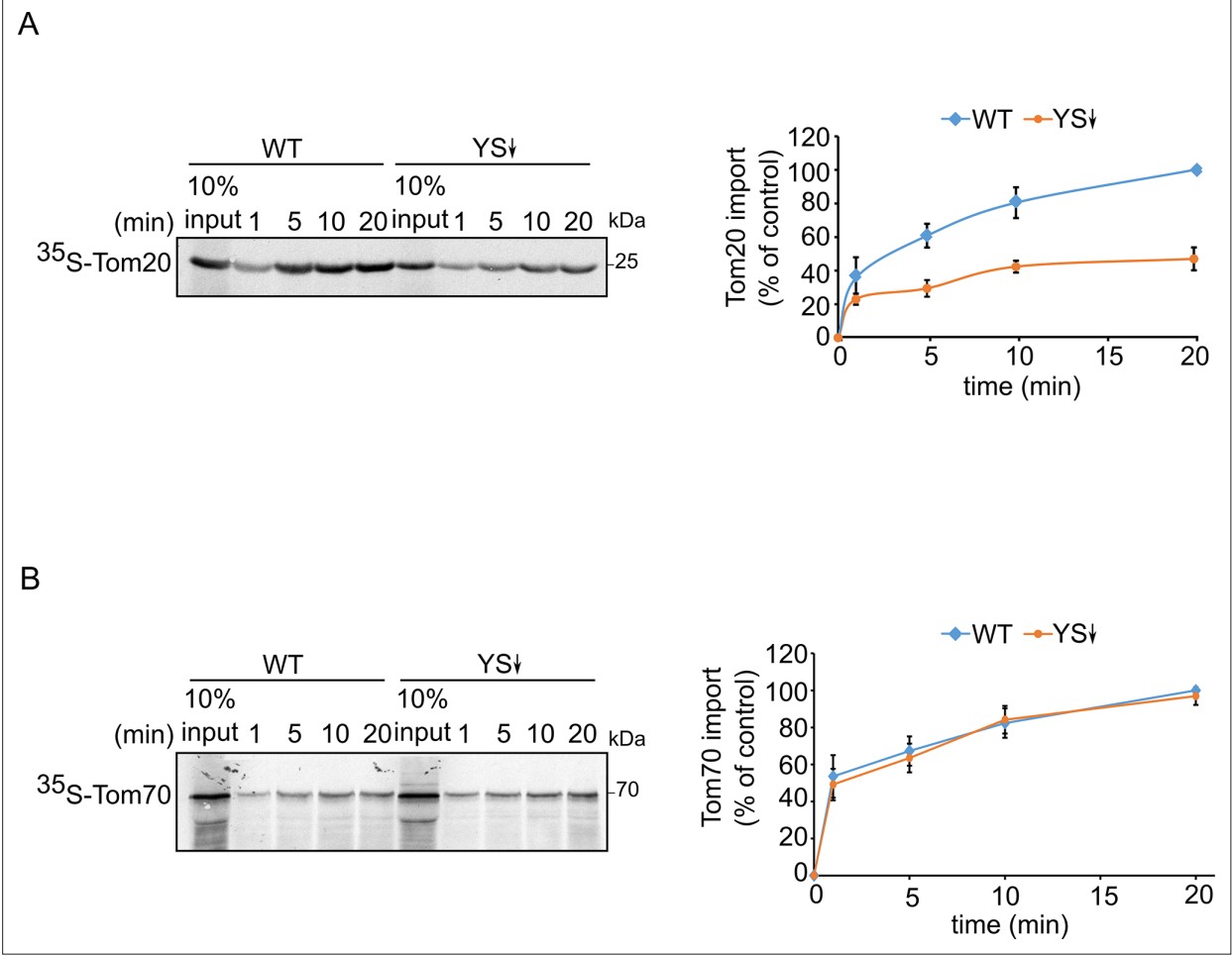

**Figure 4.** Signal-anchored proteins show variable dependence on Ydj1 and Sis1. Radiolabeled Tom20 (**A**) or Tom70 (**B**) were translated in yeast extract from either WT or Ydj1 and Sis1 depleted cells (YS↓). The radiolabeled proteins were incubated with WT mitochondria for the indicated time points (1, 5, 10, and 20 min). After import, mitochondria were subjected to alkaline extraction and the pellet was analyzed by SDS-PAGE and autoradiography. Right panels: Intensities of the bands corresponding to Tom20 and Tom70 were quantified. The intensities of the bands corresponding to import from WT yeast extract after 20 min were set to 100%. The graph represents the mean values ± SD of three independent experiments.

The online version of this article includes the following source data for figure 4:

**Source data 1.** Source data for *Figure 4A and B*.

**Source data 2.** Raw data for plot in *Figure 4A*.

**Source data 3.** Raw data for plot in *Figure 4B*.

important role in the biogenesis of SA proteins. To substantiate this assumption, we wished to examine whether inhibiting this chaperone can interfere with the import of SA proteins. Thus, we performed in vitro import assays in which the yeast extract used for protein translation was supplemented with the Hsp70 inhibitor, CBag (C-terminal Bag domain of human Bag-1M) prior to import into mitochondria (*Jores et al., 2018*). As a result, the import efficiency of both, Msp1 and Mcr1 decreased to 50% and 20%, respectively, as compared to control BSA-supplemented yeast extract (*Figure 7A and B*). We previously observed that this inhibitor does not cause a general damage to the import capacity of mitochondria as it did not affect the import of the matrix-destined protein pSu9-DHFR (*Jores et al., 2018*). Altogether, these experiments suggest a crucial role for Hsp70 in facilitating proper insertion of SA proteins into the mitochondrial OM.

Considering the physiological relevance of both Hsp70 and Hsp40 (co)chaperones for the biogenesis of SA proteins, we aimed to understand the dynamics of the complex formation between the SA substrate, the Hsp40 co-chaperones (Sis1), and the Hsp70 chaperones (Ssa1). To this end, we performed additional set of fluorescence anisotropy experiments in which Ssa1, ATP and Sis1, each a few minutes

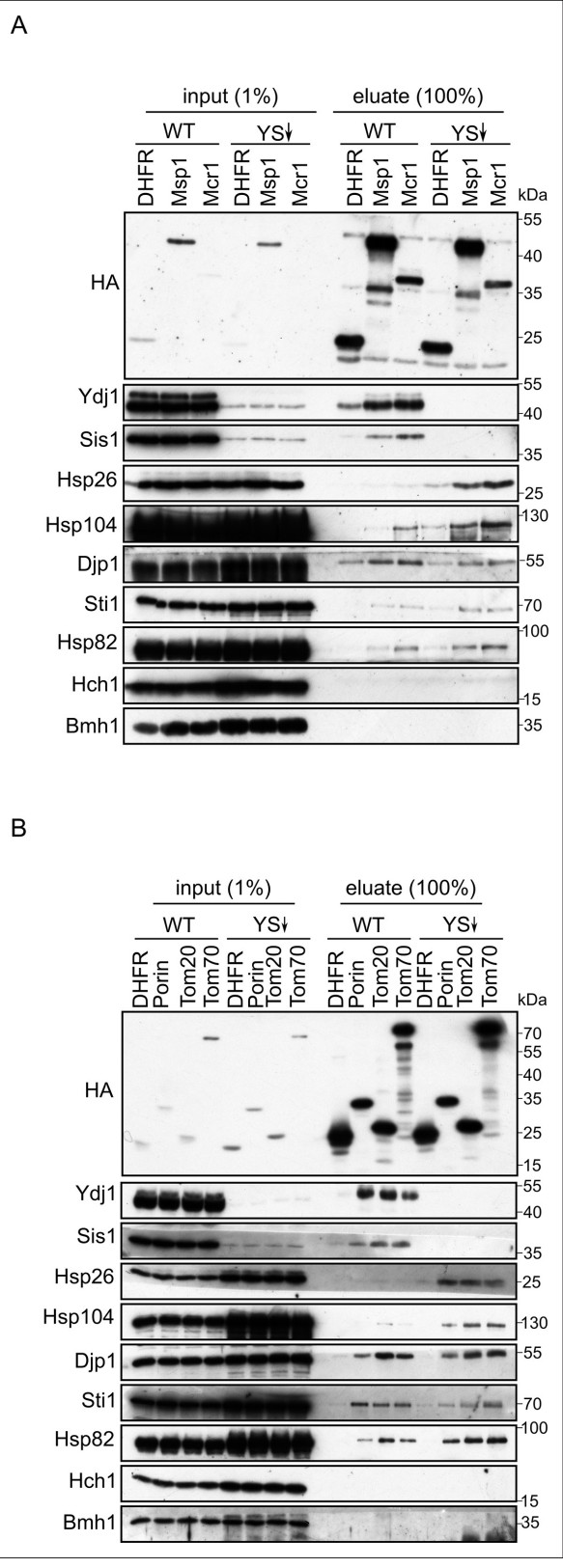

**Figure 5.** Depletion of Ydj1 and Sis1 can increase the risk for aggregation of newly synthesized proteins. In-vitro translation reactions using yeast extracts from either WT cells or from cells depleted for both Ydj1 and Sis1 (YS↓) were programmed with mRNA encoding HA-tagged versions of the indicated proteins (**A**), DHFR, Msp1, and Mcr1; (**B**), DHFR, (Porin, Tom20, and Tom70). The reactions were subjected to a pull-down with anti-HA beads. Samples

*Figure 5 continued on next page*

*Figure 5 continued*

from the input (1%) and the eluates (100%) were analyzed by SDS-PAGE and immunodecoration with the indicated antibodies.

The online version of this article includes the following source data for figure 5:

**Source data 1.** Source data for *Figure 5A*.

**Source data 2.** Source data for *Figure 5B*.

apart and in a different order were added to the Mcr1-TAMRA-labeled peptide (*Figure 7C–E*). Since Ssa1 has ATPase activity that is modulated by Hsp40 co-chaperones, we tested whether the binding of Hsp40 or Hsp70 to the substrate is ATP dependent. Higher anisotropy values were observed when Ssa1 was supplemented first, indicating complex formation between Ssa1 and Mcr1 peptide. After adding ATP, the anisotropy was reduced indicating that the complex started to disassociate as a result of ADP exchange for ATP, which promotes chaperone-substrate release. However, addition of Sis1 increased the anisotropy again suggesting either a complex formation between Sis1 and the substrate or that Sis1 restored the complex formation between Ssa1 to Mcr1 peptide by tuning the ATPase-driven chaperone cycle of Ssa1 (*Figure 7C*).

In another experiment, we added first Sis1 to the Mcr1 peptide and, as expected, complex formation was observed (*Figure 7D*). Only a slight decrease in the anisotropy was detected once Ssa1 was added to the mixture. This behavior could be attributed to a transfer of the substrate from Sis1 to Ssa1 due to higher affinity of the substrate to the latter protein. Interestingly, when ATP was supplied to the mixture at last, no further decrease in the anisotropy was measured (*Figure 7D*). This lack of effect could be explained by one of the two possibilities – (i) since there is still Sis1 in the mixture, following the Ssa1-substrate disassociation due to ATP, the substrate re-associated with Sis1, or (ii) due to the presence of Sis1, the ATP hydrolysis is accelerated which retains Hsp70 chaperone conformation in favor of substrate binding, making the Ssa1-subsatre complex no longer sensitive to ATP.

In the last experiment, Sis1 was supplemented after Ssa1 and an additional increase in anisotropy was observed (*Figure 7E*). Since the total change in anisotropy is even larger than the increase observed in the presence of either Ssa1 or Sis1 alone, this observation might suggest the formation of a ternary complex composed of Ssa1-Sis1-substrate. Importantly, this rise was not long-lasting, and the complex appears to dissociate over time (*Figure 7E*). As before, the addition of ATP at this stage did not change the level of anisotropy. Overall, these findings support a complex formation between Hsp70 and its SA substrate. In addition, they suggest a regulatory effect of ATP on this complex, which is no longer observed when Sis1 is in the mixture, suggesting a role of Sis1 in stabilizing this complex.

## Tom20 and Tom70 receptors might play an offsetting role in the biogenesis of the SA proteins Msp1 and Mcr1

Next, we wanted to decipher the late cytosolic events involving the recognition of newly synthesized proteins at the mitochondrial surface. To that goal we investigated, using different approaches, the potential involvement of the canonical import receptors Tom20 and Tom70 in the biogenesis of Mcr1 or Msp1. First, we performed in vitro pull-down assays using the cytosolic domains of either Tom20 or Tom70 N-terminally fused to GST moiety. GST alone served as a negative control. These GST-fusion proteins were incubated with freshly translated HA-tagged variants of Msp1, Mcr1 or DHFR (as a control protein). Following two hours incubation in the presence of ATP to allow release of potential bound chaperones, the elution fraction was collected, and bound proteins were detected using anti-HA antibody. Msp1 and Mcr1 displayed variable degrees of interaction with both receptors. After normalizing for the background binding to GST, Mcr1 exhibited four times higher binding to the cytosolic domain of Tom20 compared to that of Tom70, while Msp1 interacted with both almost equally. Of note, no binding was detected when DHFR was incubated with the receptors, demonstrating the specificity of the assay (*Figure 8A*).

Since prior to their incubation with the receptors, the SA proteins were translated in yeast extract containing the repertoire of molecular chaperones, we asked whether chaperones can be involved in the recognition process of the newly synthesized proteins by the receptors. Hence, we aimed to examine the potential interaction between the TOM receptors and different chaperones using the same approach, namely, pull-down from yeast extract. Notably, Ydj1 showed similar binding levels

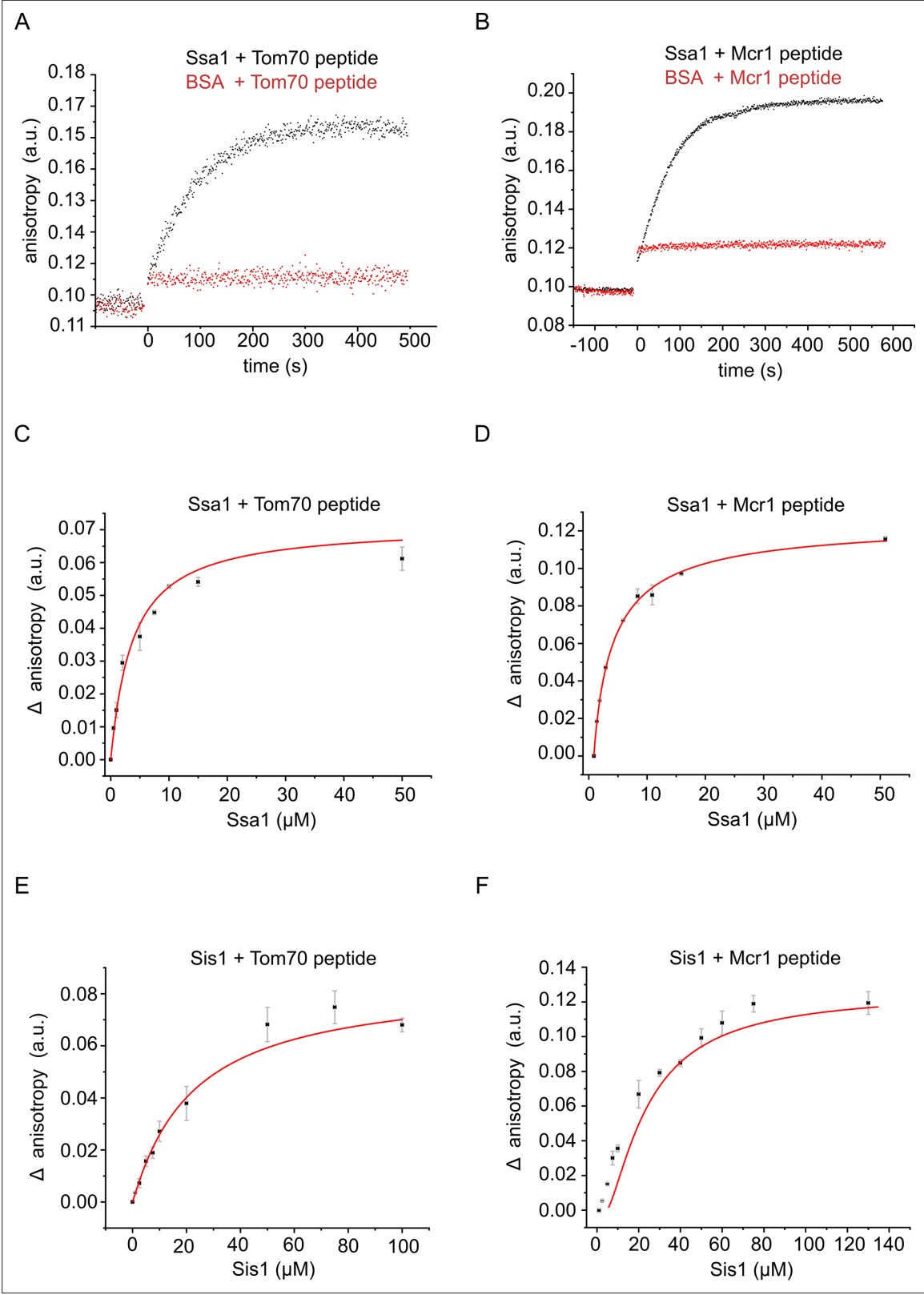

**Figure 6.** The hydrophobic segment of the signal-anchored proteins interacts with the Hsp70 chaperone and its co-chaperone Sis1. (**A and B**) The fluorescence anisotropy of TAMRA-labeled peptides corresponding to the TMSs of either Tom70 (**A**) or Mcr1 (**B**) was measured in the presence of 10 µM of the Hsp70 Ssa1 (black circles) or 30 µM BSA, as a control (red circles). (**C–F**) For affinity determinations, the TMS-labelled peptides of either Tom70 (**C and E**) or Mcr1 (**D and F**) were mixed with the indicated concentrations of either Ssa1 (**C and D**) or Sis1 (**E and F**), and the difference in anisotropy (Δ anisotropy) between the bound and free peptide was plotted against the (co)chaperone concentrations.

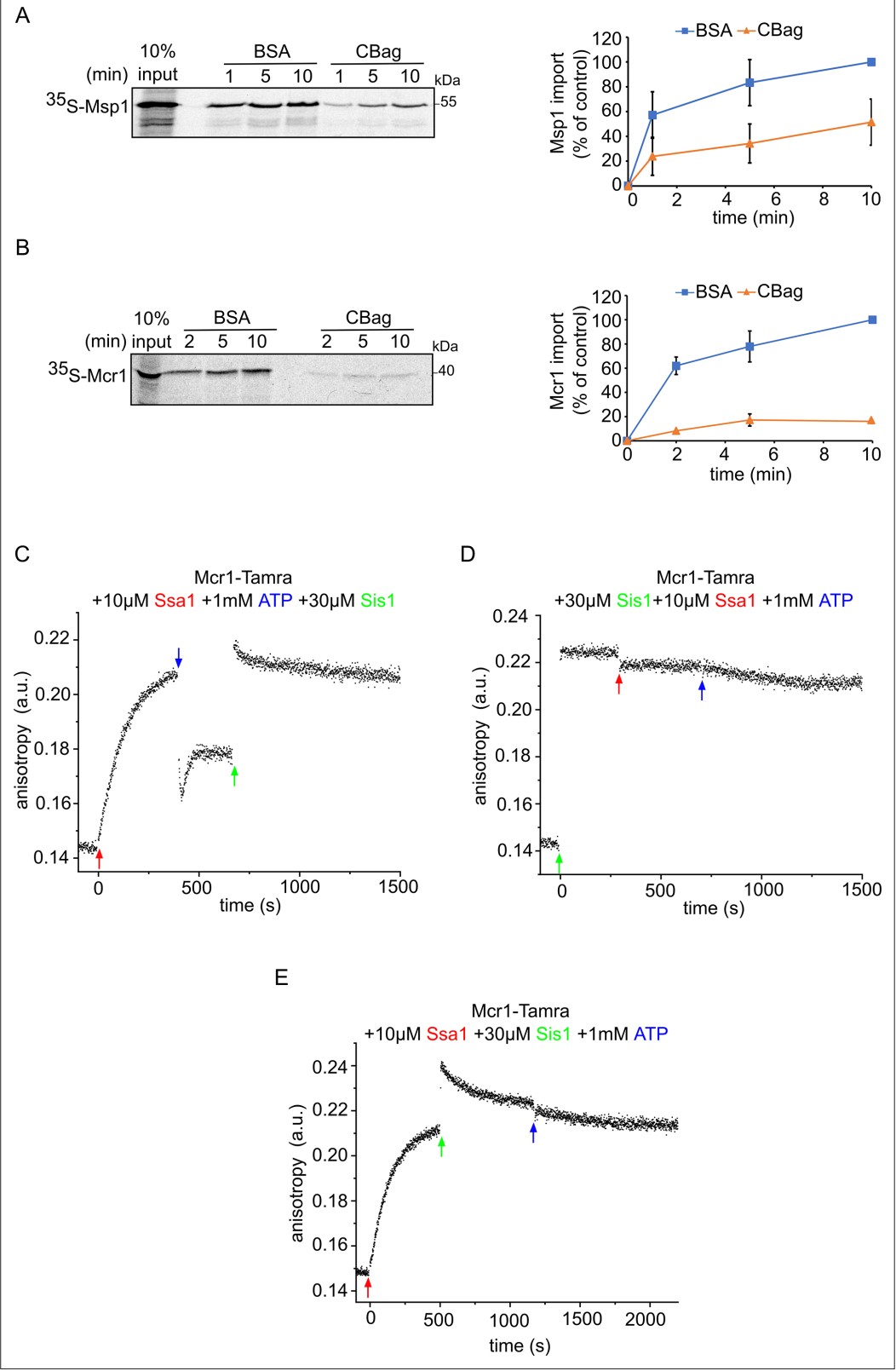

**Figure 7.** The Hsp70 chaperone Ssa1 is required for proper membrane integration of signal-anchored proteins. (**A and B**) Left panels: Radiolabeled Msp1 (**A**) and Mcr1 (**B**) were translated in yeast extract from WT cells and subjected to in-vitro import assay using isolated mitochondria. Prior to the import, the yeast extract translation reaction was incubated with either CBag (Hsp70 inhibitor) or with BSA, as a control. After import for the indicated

*Figure 7 continued on next page*

*Figure 7 continued*

time periods, the samples were subjected to carbonate extraction and the pellets fraction were analysed by SDS-PAGE followed by autoradiography. Right panels: The bands corresponding to Msp1 and Mcr1 were quantified and the results of three independent experiments are presented as mean values ± SD. The intensities of the bands corresponding to import for 10 min in the presence of BSA were set to 100%. (**C–E**) The fluorescence anisotropy of TAMRA-labelled Mcr1-TMS peptide was measured while supplementing 10 µM Ssa1, 30 µM Sis1, and 1 mM ATP in the order indicated in the various panels.

The online version of this article includes the following source data for figure 7:

**Source data 1.** Source data for *Figure 7A and B*.

**Source data 2.** Raw data for plot in *Figure 7A*.

**Source data 3.** Raw data for plot in *Figure 7B*.

to both receptors while Ssa1/2, Sti1 and Hsp104 exhibited stronger binding to the cytosolic domain of Tom70. On the other hand, Aha1 displayed a stronger interaction with the Tom20 receptor. Other chaperones, like Hsp26 and Hch1 did not bind to either receptor (*Figure 8—figure supplement 1A*).

Since the cooperation between Hsp70 chaperone and Tom70/Tom20 receptors in facilitating productive delivery of a subset of preproteins to the receptor for subsequent membrane translocation has been reported in multiple publications (*Mills et al., 2009*; *Fan et al., 2010*; *Fan et al., 2011*; *Komiya et al., 1997*; *Young et al., 2003*; *Wegele et al., 2006*), we decided to further investigate the implications of such interactions for the biogenesis of SA proteins. Given that Tom70 contains docking site for Hsp70 and Hsp90 chaperones, along with our observation that Ssa1/2 is probably involved in the insertion process of Msp1 and Mcr1, we asked whether Tom70 receptor recognizes the polypeptide substrate while in complex with Ssa1 and whether Hsp70 docking is required for the formation of a productive substrate/Tom70 complex. For this purpose, we analyzed Mcr1-peptide binding to GST-Tom70 receptor in the presence of Ssa1, Sis1, and ATP supplemented in different orders and monitored the change in the anisotropy. As shown in the previous experiments (*Figure 7C*), we observed complex formation after adding Ssa1 to Mcr1-peptide, which was detached upon ATP addition. However, once GST-Tom70 was added, a complex that caused an even higher anisotropy shift was formed (*Figure 8B*), suggesting that Mcr1 peptide, after being released from Ssa1, formed a complex with Tom70 with higher affinity. To exclude unspecific binding that might form between the GST moiety and the substrate, GST alone or GST-Tom70 were added to Mcr1-peptide. Whereas no change in anisotropy was observed when GST alone was used, high anisotropy values were detected following addition of GST-Tom70 (*Figure 8—figure supplement 1B*). Interestingly, a complex between Mcr1-peptide and Tom70, as indicated by a large increase in the measured anisotropy, was also formed when GST-Tom70 was supplemented first while Ssa1 was still absence (*Figure 8C*). Once Ssa1 and ATP were supplemented to this Tom70-Mcr1(TMS) complex, only a slight reduction shift was observed (*Figure 8C*), suggesting that the complex between Mcr1 substrate and Tom70 receptor remained mostly stable. This observation supports the assumption that the substrate interacts with the receptor with higher binding affinity than to the chaperone. Such an increased affinity can promote transfer of the substrate from the chaperone to the receptor.

To test if the substrate-receptor interaction can occur even when the substrate is compounded with the Ssa1 chaperone, we enabled substrate-chaperone complex to form by supplementing Ssa1 first, followed by the addition of GST-Tom70. Surprisingly, higher anisotropy shift was detected right after addition of GST-Tom70, indicating that the Tom70 receptor can bind the substrate while the latter is in complex with the chaperone. Addition of ATP, which aids the release of the substrate from the chaperone, resulted in an even higher increase in anisotropy (*Figure 8D*), suggesting that the Tom70 receptor might bind the substrate with enhanced efficiency upon its release from the chaperone. To ensure that the receptor can indeed bind the substrate-chaperone complex, we stabilized the complex formed by Ssa1-Mcr1(TMS) by co-addition of Sis1 to the mixture. Even in this case, the receptor Tom70 still interacted with the substrate (*Figure 8E*). Overall, these findings suggest that the substrate has higher affinity for the receptor than to the chaperone. Furthermore, Tom70 is capable of binding substrates irrespective of the presence or absence of the Hsp70 chaperone Ssa1.

Having demonstrated that Tom20 and Tom70 can bind newly translated SA proteins to a different degree (*Figure 8A*), we aimed to assess whether these interactions are required for their mitochondrial insertion. Previous studies have shown that the absence of either one of the TOM receptors does

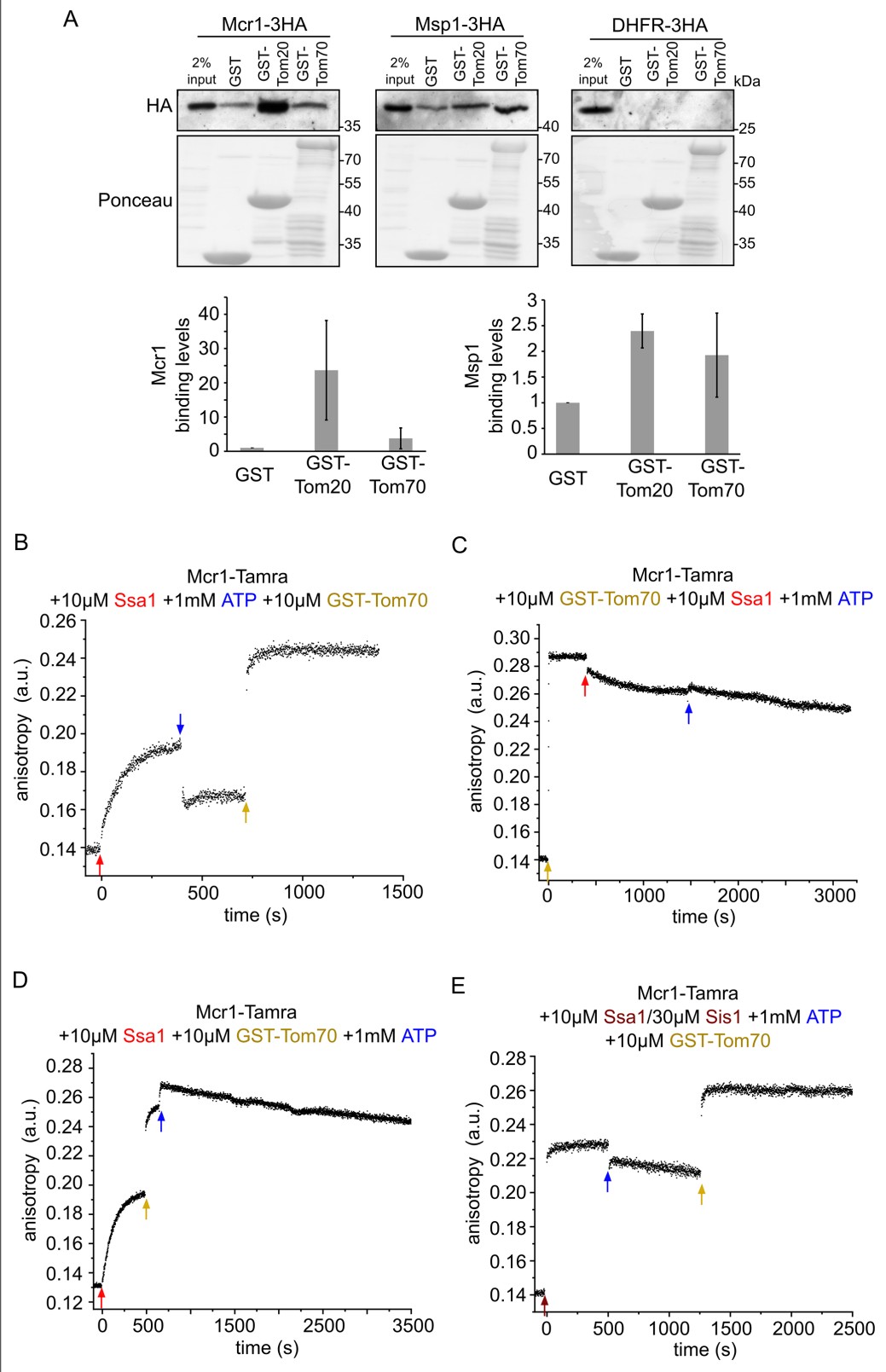

**Figure 8.** Newly synthesized signal-anchored proteins can be recognized by the cytosolic domains of the TOM receptors. (**A**) HA-tagged versions of the signal-anchored proteins Mcr1 and Msp1 or of the control protein DHFR were freshly translated in yeast extract. Next, the newly translated proteins were mixed with GST alone or GST fused to the cytosolic domain of either Tom20 (GST-Tom20) or Tom70 (GST-Tom70) bound to glutathione

*Figure 8 continued on next page*

*Figure 8 continued*

beads. Input (2%) and eluate (100%) samples were subjected to SDS-PAGE. GST fusion proteins were detected by Ponceau staining whereas the HA-tagged proteins via immunodecoration against the HA-tag. Lower panels: Bands corresponding to Msp1-3HA and Mcr1-3HA from three independent experiments were quantified and the level of binding to GST alone was set as 1. Error bars represent ± SD. (**B–D**) Fluorescence anisotropy of TAMRA-labeled Mcr1 peptide was monitored after supplementing the reaction with 10 µM Ssa1, 1 mM ATP, or 10 µM GST-Tom70 in the indicated order. (**E**) As in panels B-D while the first addition was of 10 µM Ssa1 together with 30 µM Sis1, followed by addition of 1 mM ATP and then finally 10 µM GST-Tom70.

The online version of this article includes the following source data and figure supplement(s) for figure 8:

**Source data 1.** Source data for *Figure 8A*.

**Source data 2.** Raw data for plots in *Figure 8A*.

**Figure supplement 1.** The TOM receptors interact with various chaperones.

**Figure supplement 1—source data 1.** Source data for *Figure 8—figure supplement 1*.

not cause a reduction in the mitochondrial steady state levels of Msp1 or Mcr1 (*Vitali et al., 2020*; *Meineke et al., 2008*). In line with these previous findings, the insertion efficiency of radiolabeled Msp1 or Mcr1, as observed in in vitro import assays, was not significantly reduced when isolated mitochondria lacking Tom70/71 were used (*Figure 9—figure supplement 1A*, B). Since both receptors, Tom20 and Tom70 can interact with Msp1 and Mcr1, we assume that both might play a role in vivo in their recognition, hence, when one receptor is absent, the other one can compensate due to their partial overlapping function (*Young et al., 2003*). In support of this inference, a slight increase in the steady state levels of either Tom20 or Tom70 was detected in cells deleted of the other receptor (*Figure 9—figure supplement 1C*).

To further test the involvement of these import receptors, we wanted to examine the direct effect of the loss of both receptors on the mitochondrial insertion of Msp1 and Mcr1. Since the double deletion of both receptors is lethal, we searched for other means how to deactivate both receptors at once. To this end, we conducted in vitro import assays of radiolabeled proteins (Msp1 and Mcr1) under two different conditions. In the first set of experiments, mitochondria were pre-treated with proteases (trypsin or proteinase K) to remove exposed domains of import receptors. In the other set of assays we employed mitochondria lacking the Tom20 receptor (isolated from *tom20Δ* strain) and treated with the C-terminal domain of human Hsp90 (C90) which is known to block the Hsp70 chaperone binding site on the mitochondrial import receptor Tom70, thus inhibiting its activity (*Young et al., 2003*). Following protease treatment of control isolated mitochondria, the cytosolic domains of both Tom20 and Tom70 were degraded, and the native proteins could not be detected by western blotting (*Figure 9A and B*). While membranal insertion of both Msp1 and Mcr1 was drastically reduced upon using trypsin-treated mitochondria, only much milder reduction was observed following PK-treatment (*Figure 9A and B*). This observation can be explained by the different cleavage sites specificity of both proteases. Since the cytosolic domains of import receptors are tightly folded, we assume that while the positive charged Lys and Arg, which are recognized by trypsin, are available on exposed segments, the hydrophobic residues, that are preferentially recognized by PK, are buried within the folded structure. Hence, trypsin might cleave larger parts of these proteins.

In the other approach, when WT mitochondria was treated with C90, which results in blockage of Tom70, import of Msp1 was reduced by 20%. This seemingly contradicts former study in which mitochondrial levels of Msp1 were not decreased in *TOM70* deleted strain (*Vitali et al., 2020*). However, this difference can be explained by the fact that in the previous study, the cells lacking Tom70 could adapt for its loss, while the effect of inhibition by C90 is immediate and not reversible. Interestingly, insertion efficiency of Msp1 into mitochondria lacking Tom20 was around 25% higher as compared to control WT mitochondria, maybe because of the slightly elevated levels of Tom70 in these cells (*Figure 9—figure supplement 1C*). Supporting this assumption, when the samples lacking Tom20 were treated with C90, the membrane integration of Msp1 was reduced by 32% (*Figure 9C*). Similar results were obtained with Mcr1 (*Figure 9D*). These findings substantiate the importance of Tom70 as a docking site for substrates SA proteins associated with (co)chaperones.

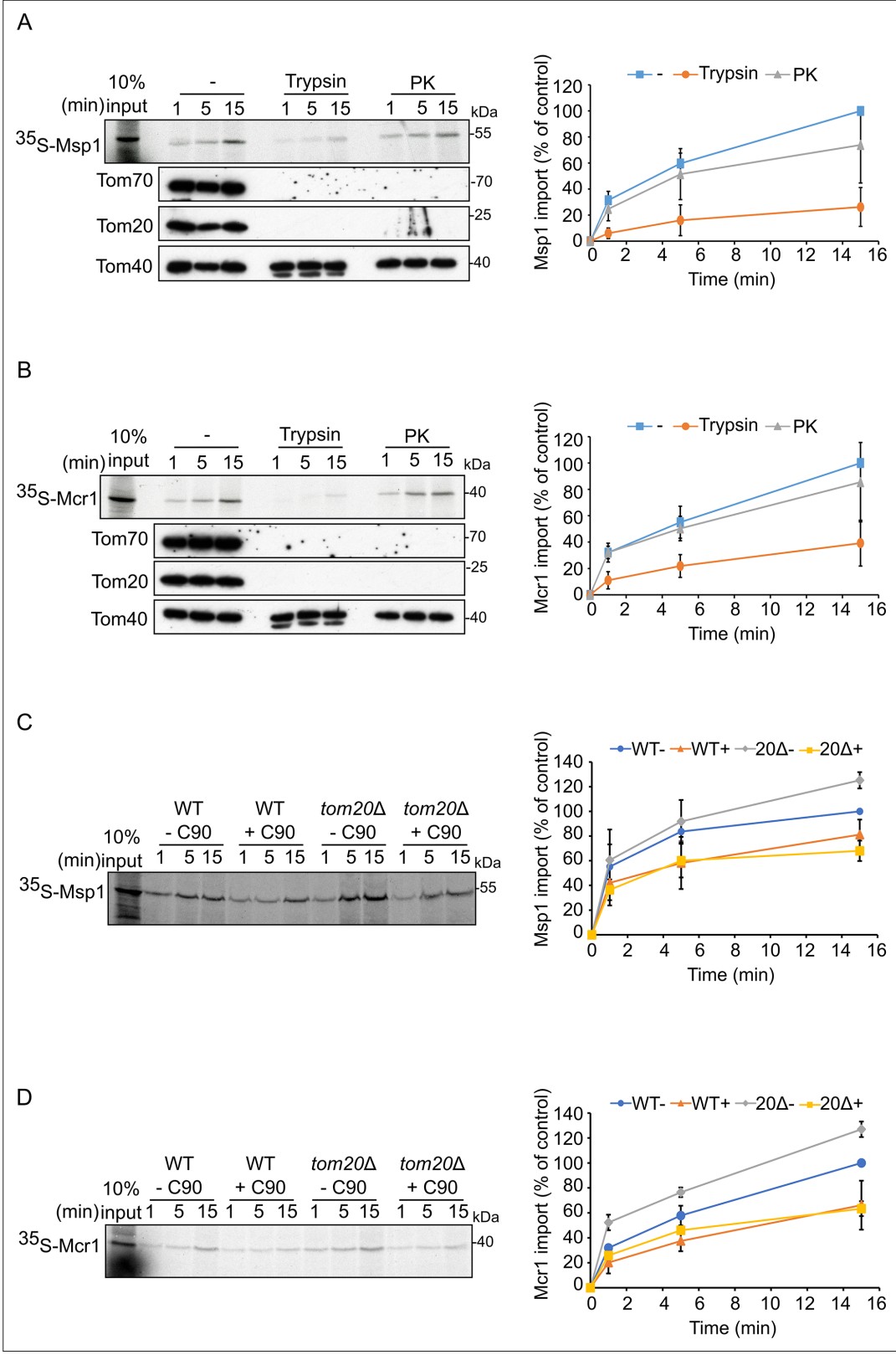

**Figure 9.** Tom70 and Tom20 may have offsetting function in mediating the biogenesis of Msp1 and Mcr1. (**A and B**) Left panels: Radiolabeled Msp1 (**A**) and Mcr1 (**B**) were translated in yeast extract from WT cells and subjected to in vitro import assay using isolated mitochondria. Prior to the import reactions, isolated mitochondria were incubated for 30 min in the presence or absence of either trypsin or proteinase K (PK). After import for

*Figure 9 continued on next page*

*Figure 9 continued*

the indicated time periods, the samples were subjected to carbonate extraction and the pellet fractions were subjected to SDS-PAGE followed by autoradiography. To verify the activity of the proteases, the same membranes were immunodecorated with antibodies against the indicated proteins. Right panels: The bands corresponding to Msp1 or Mcr1 were quantified and the results of three independent experiments are presented as mean values ± SD. The intensities of the bands corresponding to import for 15 min in the absence of protease were set to 100%. (**C and D**) Left panels: Radiolabeled Msp1 (**C**) and Mcr1 (**D**) were translated in yeast extract from WT cells and subjected to in-vitro import assay using mitochondria isolated from either WT or *tom20Δ* cells. Prior to the import reactions, mitochondria were incubated in the presence or absence of 20 µM C90 (blocker of Tom70). After import for the indicated time points, the samples were subjected to carbonate extraction and the pellet fractions were analyzed by SDS-PAGE followed by autoradiography. Right panels: The bands corresponding to Msp1 or Mcr1 were quantified and the results of three independent experiments are presented as mean values ± SD. The intensities of the bands corresponding to import for 15 min in the absence of C90 were set to 100%.

The online version of this article includes the following source data and figure supplement(s) for figure 9:

**Source data 1.** Source data for *Figure 9A and B*.

**Source data 2.** Source data for *Figure 9C and D*.

**Source data 3.** Raw data for plots in *Figure 9A and B*.

**Source data 4.** Raw data for plots in *Figure 9C and D*.

**Figure supplement 1.** Biogenesis of SA proteins is not affected by the single deletion of a TOM receptor.

**Figure supplement 1—source data 1.** Source data for *Figure 9—figure supplement 1A and B*.

**Figure supplement 1—source data 2.** Source data for *Figure 9—figure supplement 1C*.

## Discussion

The early cytosolic events in the biogenesis of mitochondrial outer membrane proteins are believed to involve cytosolic factors and chaperones to keep the newly synthesized proteins in an import competent conformation, which is crucial for their effective membranal insertion. Such factors have been identified for some of the mitochondrial outer membrane proteins, such as the β-barrel proteins (*Jores et al., 2018*), and some single-span proteins (*Cichocki et al., 2018*; *Papić et al., 2013*; *Opaliński et al., 2018*). In this study, we employed four different model proteins from the SA family and identified several cytosolic chaperones that can interact with such newly synthesized proteins through their hydrophobic transmembrane domain. Of note, the cytosolic domains of these SA proteins were not associated with chaperones indicating that they are less prone to forming aggregates or to misfold in the cytosol. These observations are in line with the widely accepted concept that chaperones recognize mainly hydrophobic patches to bind their substrates (*Li et al., 2009*; *Saio et al., 2014*; *Clerico et al., 2015*).

We identify chaperones from the Hsp90, Hsp70, and Hsp40 families to interact with SA proteins. When we tested the role of the Hsp40 co-chaperones, Ydj1 and Sis1, which are two major J-domain proteins in the yeast cytosol, we observed that mitochondrial steady state levels of Tom20 and Tom70 were largely reduced upon depletion of both chaperones and this effect was less pronounced when only one of the co-chaperones was depleted. This observation indicates, in agreement with an earlier study (*Johnson and Craig, 2001*), that Ydj1 and Sis1 share overlapping functions. In contrast, the steady-state levels of Msp1 and Mcr1, despite exhibiting interaction with Ydj1 and Sis1, were unaffected or only slightly affected, respectively , by the co-depletion of the co-chaperones. This suggests that other Hsp40 co-chaperones play most likely a more central role in the biogenesis of these SA proteins. Furthermore, we could demonstrate that upon co-depletion of both Ydj1 and Sis1, the newly synthesized SA proteins showed enhanced binding to Hsp26 and Hsp104 chaperones that are implicated in binding aggregated proteins (*Zolkiewski et al., 2012*; *Zhou et al., 2011*; *Franzmann et al., 2005*). These findings emphasize the involvement of Ydj1 and Sis1 in preventing cytotoxic protein aggregation and agree with other proposed functions of both co-chaperones (*Cyr, 1995*; *Klaips et al., 2020*).

Additionally, we have shown that both the in vitro import efficiency and the steady-state levels of Tom20 are reduced upon depletion of Ydj1 and Sis1. In the case of Tom70, in vitro insertion was unaffected despite reduced mitochondrial steady state levels, suggesting that such chaperones might play a role that is upstream of Tom70 membranal insertion. These combined observations suggest that

Ydj1 and Sis1 are involved in the biogenesis of SA proteins to varying degrees. The cis characteristics of the substrate proteins that dictates the variable dependency will be the topic of future studies.

We also validated via fluorescence anisotropy the interactions between a peptide corresponding to the transmembrane of Mcr1 and the two (co)chaperones Ssa1 and Sis1. Interestingly, the binding affinity of the substrate to the co-chaperone (Sis1) was 10-fold lower than the binding to the Hsp70 chaperone (Ssa1). These observations support the hypothesis that the substrate transfer from the co-chaperone to the main Hsp70 chaperones is driven by increased affinity to the latter. Similar results were also previously observed for a peptide representing the mitochondrial targeting element of the β-barrel protein Porin (*Jores et al., 2018*). According to our findings, the complex formed by Ssa1/substrate is regulated by the J-protein Sis1. Although we could show that a potential substrate is able to bind Hsp70 chaperone also in the absence of Sis1, this interaction was susceptible to ATP. In contrast, in the presence of Sis1 the complex was no longer responsive to ATP, pointing out the regulatory function of Sis1 on the Hsp70 ATPase activity. In agreement with our findings, previous studies have shown that both Sis1 and Ydj1 facilitate binding between substrate and Hsp70 chaperone (*Kampinga and Craig, 2010*).

Our current findings show that inhibiting Ssa1 activity significantly reduces the integration of Msp1 and Mcr1 into the OM. This implies that in addition to its well-known function in facilitating protein folding, Ssa1 also directly supports the mitochondrial insertion of Msp1 and Mcr1. Of note, we could show that SA proteins can interact with the TOM receptors Tom70 and Tom20, suggesting that TOM receptors may play a role in recognizing newly synthesized SA proteins on the mitochondrial surface. At least for Mcr1, such a recognition by the import receptors might be related to the presence of MTS-like stretch in residues 1–10 of the protein (*Hahne et al., 1994*). Furthermore, in accordance with previous observations (*Backes et al., 2021*; *Kreimendahl and Rassow, 2020*; *Young et al., 2003*), this recognition may involve an interplay between the Hsp70 chaperone and the Tom70 receptor. Based on our findings, we propose that Ssa1/substrate complex is initially identified by Tom70 receptor, most likely using the docking site in Tom70, which may be crucial for correct targeting.

However, it is important to note that based on crystal structure, the cytosolic segment of Tom70 receptor contains two distinctive domains, the C-terminal domain which acts as substrate binding pocket and N-terminal domain (clamp domain) consisting of three TPR motifs, which is most likely responsible for binding the EEVD motif of Hsp70 and Hsp90 chaperones (*Wu and Sha, 2006*). Consonant with that, we assume that in our in vitro assays, Tom70 was capable of binding both, the Ssa1/substrate complex, likely via the N-terminal chaperone docking domain and the substrate alone following its release from the chaperone through its C-terminal substrate binding pocket. The substrate is then released from the chaperone-complex and relayed to Tom70 to which it has a higher affinity. These findings support a previously published study which proposed that the monomeric form of Tom70 is responsible for mediating initial chaperone docking and precursor recognition via its clamp domain followed by substrate release. This recognition is assumed to be facilitated by the exchange of ATP for ADP at the chaperone ATPase domain and results in the dimerization of Tom70 which favors interactions solely with the substrate (*Mills et al., 2009*).

When we analyzed the direct involvement of the TOM receptors in the biogenesis of Msp1 and Mcr1, we found that their insertion was drastically reduced upon either trypsin-mediated removal of the exposed domains of surface proteins or inhibition of Tom70 by C90. Of note, the membrane integration of these substrates was enhanced upon *TOM20* deletion. This finding is in line with previous study showing higher steady state levels of Msp1 in mitochondria devoid of Tom20 (*Vitali et al., 2020*). This enhancement could be attributed to higher expression of Tom70 to compensate for the deletion of Tom20. Moreover, inhibition of Tom70 in a deletion strain of Tom20 led to further reduction in the insertion of SA proteins, supporting the assumption that each receptor can compensate for the absence of the other one. Hence, it seems that both receptors might be involved in the biogenesis of the SA proteins Msp1 and Mcr1. Yet, it is still unclear whether such proteins might follow different insertion pathways based on the receptor available for their recognition, specially that Tom20 does not have a chaperone docking site but was rather suggested to have a chaperone-like activity to prevent protein aggregation (*Yano et al., 2004*).

Altogether, our current findings, when combined with prior information, provide new insights into the cytosolic chain of events from the synthesis of SA proteins until their recognition at the mitochondrial surface allowing us to suggest a working model for their biogenesis (*Figure 10*). Following

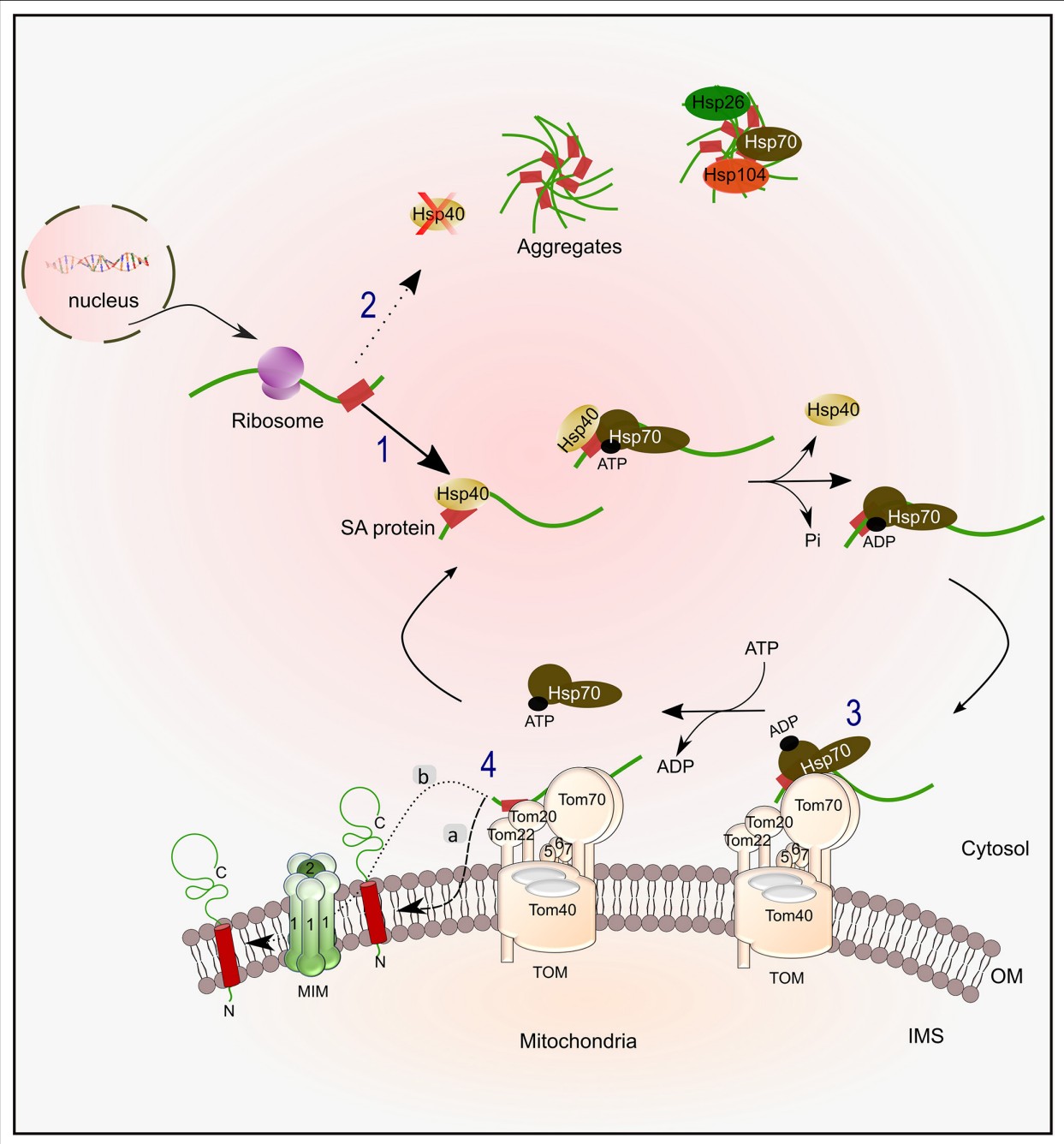

**Figure 10.** Working model for the biogenesis of SA proteins. After SA proteins get synthesized on cytosolic ribosomes, they can associate with Hsp40 chaperones (like Ydj1 and Sis1)(1). Upon depletion of Hsp40 chaperones, newly synthesized SA proteins might tend to form aggregates, which can then associate with disaggregases chaperones such as Hsp104 and Hsp26 (2). Hsp40 chaperones drive the transfer of the newly synthesized protein to Hsp70 chaperone (Ssa1/2) by facilitating the conversion of Hsp70 from its ATP form to the ADP one that has a higher affinity for polypeptides. Next, the protein-chaperone complex is recognized by Tom70 receptor (3), followed by disassociation of the chaperone. Subsequently to the recognition by Tom70, which may involve also Tom20, the substrate is then inserted into the OM, either through an unassisted route (4 a), or via a pathway which is facilitated by the MIM complex (4b).

their synthesis, SA proteins interact with molecular chaperones via their hydrophobic patches (*Figure 10*, stage 1). This productive binding protects the newly synthesized proteins from aggregation (*Figure 10*, stage 2). Hsp40 cochaperone initially bind such proteins and facilitate their transfer to the Hsp70 chaperone by promoting ATP hydrolysis resulting in Hsp70-ADP bound state that has higher affinity to the substrate. Such interactions are not only crucial for keeping the substrate protein

in an import-competent conformation by preventing aggregation and misfolding events, but also for their proper mitochondrial targeting which involves the TOM complex receptors Tom20 and Tom70 (*Figure 10*, stage 3). Binding of the chaperone/substrate complex to Tom70 receptor might result in disassociation of the substrate from the Hsp70 chaperone leading to its ultimate insertion into the OM that is usually facilitated by the MIM complex (*Figure 10*, stage 4). Although both receptors may not be vividly crucial for the actual membrane integration process, we propose that they are required for facilitating efficient delivery in the crowded environment of the cell.

## Materials and methods

### Yeast strains and growth methods

Yeast strains used in the study were isogenic to *Saccharomyces cerevisiae* strains W303α or JSY7452. The parental strain YMK120 was used to create strains with genes under the control of a tetracycline-repressible promoter (*Gnanasundram and Koš, 2015*). The tetracycline operator was inserted into the genome by homologous recombination using an insertion cassette amplified from the plasmids pMK632Kan and pMK632His (*Gnanasundram and Koš, 2015*). Strains with two genes under the control of the tetracycline operator were obtained by mating of strains with a single tet-regulated gene followed by tetrad dissection. Yeast cells were grown in lactate (2% [w/v]) as carbon source for mitochondria isolation. Yeast strains expressing Ydj1 and Sis1 under tetracycline promotor were grown in YP-Sucrose media (for mitochondria isolation) or in glucose-containing rich medium, YPD (for isolation of cell extract). For mitochondria isolation, cells were grown for 2 hr on medium lacking doxycycline before supplementing them with 2 µg/ml of doxycycline. Then, cells were cultured for different time periods before their harvest. All strains used in this study are listed in key resources table.

### Recombinant DNA techniques

The plasmids pRS426-TPI-Tom20 and pRS426-TPI-Tom70 were used as templates for the PCR-amplification of the *TOM20* and *TOM70* genes, respectively. The amplification product was inserted into the plasmid pGEM4polyA-3HA using KpnI and BamHI restriction sites. The *MSP1* gene was amplified by PCR from yeast genomic DNA with specific primers containing BamHI and HindIII restriction sites. The PCR product was cloned into the plasmid pGEM4 to obtain pGEM4-yk-Msp1 which was used as a template for cloning yk-Msp1 into pGEM4polyA-3HA plasmid using SacI and BamHI restriction sites. The *MCR1* gene was amplified by PCR using pGEM4-yk-Mcr1 plasmid as template. The amplification product was inserted into pGEM4polyA-3HA using EcoRI and SmaI restriction sites.

The sequences encoding either the cytosolic domain (a.a 33–363) or the transmembrane domain (a.a 1–32) of Msp1 were amplified by PCR using the pGEM4-YK-Msp1-3HA plasmid as a template. The sequence encoding either the cytosolic domain (a.a 35–302) or the transmembrane domain (a.a 1–39) of Mcr1 were amplified by PCR using pGEM4-YK-Mcr1-3HA plasmid as a template. The sequences encoding either the cytosolic domain (a.a 33–183) or the transmembrane domain (a.a 1–30) of Tom20 were amplified by PCR using the pGEM4-YK-Tom20-3HA plasmid as a template. The sequences encoding either the cytosolic domain (a.a 33–617) or the transmembrane domain (a.a 1–32) of Tom70 were amplified by PCR using the pGEM4-YK-Tom70-3HA plasmid as a template. All these PCR products were inserted with suitable restriction sites into the pGEM4polyA-3HA plasmid for their translation in a yeast extract. In all the constructs mentioned above the yeast Kozak (YK) sequence was introduced via a primer directly upstream of the start codon. All primers and plasmids used in this study are listed in key resources table.

### Protein purification

Recombinant cBag and C90 were expressed in BL21 cells from the plasmid pPROEX-HTa-cBag or pPROEX-HTa-C90, respectively (*Young et al., 2003*). Expression was induced for 4 hr with 1 mM IPTG at 37 °C. The cells were harvested, resuspended in French Press buffer (40 mM HEPES, 100 mM KCl, 20 mM imidazole, 2 mM PMSF, EDTA-free cOmplete protease inhibitor [Roche], pH 7.5) for 1 hr followed by homogenization using tight douncer. Cells were then lysed with an EmulsiFlex-C5 French Press. The cell lysate was subjected to a clarifying spin (15,000 x g, 15 min, 4 °C) and the supernatant was incubated overnight with 2 mL Ni-NTA Agarose beads (Cube Biotech). The bound proteins were

washed with 20 mL wash buffer (40 mM HEPES, 100 mM KCl, 50 mM imidazole, pH 7.5) and eluted with elution buffer (40 mM HEPES, 100 mM KCl, 300 mM imidazole, pH 7.5).

GST, GST-Tom70 and GST-Tom20 were expressed and purified as described earlier (*Papić et al., 2013*).

## Biochemical procedures

Protein samples for immunodecoration were analyzed on 8, 12, or 15% SDS-PAGE and subsequently transferred onto nitrocellulose membranes by semi-dry western blotting. Proteins were detected by incubating the membranes first with primary antibodies and then with horseradish peroxidase-conjugates of either goat anti-rabbit or goat anti-rat secondary antibodies. A list of the antibodies used in this study is found in key resources table.

Mitochondria were isolated from WT yeast cells for in vitro imports by differential centrifugation as described. Pull-down assays with in vitro translated proteins were performed using cell-free yeast extract as described before (*Jores et al., 2018*).

## Blue native PAGE

Mitochondria (100 µg) were lysed in 50 µl buffer containing digitonin (1% digitonin, 20 mM Tris–HCl, 0.1 mM EDTA, 50 mM NaCl, 10% (v/v) glycerol, 1 mM PMSF, pH 7.2). After incubation on ice for 30 min and clarifying spin (30,000xg, 20 min, 2 °C), 5 µl of sample buffer (5% (w/v) Coomassie blue G, 500 mM 6-amino-N-caproic acid, 100 mM Bis-Tris, pH 7.0) were added, and the mixture was analyzed by electrophoresis on a 4–12% gradient BN-PAGE. Gels were blotted on polyvinylidene fluoride membranes, and proteins were further analyzed by immunodecoration.

## In vitro translation and GST Pull-down

Purified GST and GST-fusion proteins (GST-Tom20 and GST-Tom70) were incubated with glutathione sepharose beads for 1 hr followed by 1 hr blocking with 5% BSA in GST basic buffer (20 mM Hepes, 100 mM NaCl, 1 mM MgCl$_2$, protease inhibitor mix, pH 7.25). The beads with the bound GST-proteins were centrifuged (500xg, 1 min, 2 °C). In vitro translation reactions containing HA-tagged proteins were centrifuged (100,000xg, 60 min, 2 °C) to remove ribosomes and aggregated proteins and were then mixed with the beads for 1 hr. An aliquot of 2% of the translated material was taken as input. The reaction was supplemented with ATP every 30 min. The beads were then washed three times with GST basic buffer and the bound material was eluted by incubating the beads for 10 min at 95 °C in 4 x sample buffer supplemented with 0.05% β-ME. Eluted material was analyzed by SDS-PAGE followed by Ponceau staining and western blotting using antibody against either the indicated proteins or the HA tag.

## In vitro translation and import of radiolabeled proteins

Yeast extracts for in vitro translation were prepared as described (*Wu and Sachs, 2014*). For the preparation of yeast extracts from Ydj1- and Sis1-depleted cells, cells were grown for 8 hr in the presence of 2 µg/mL doxycycline prior to extract preparation.

Proteins were synthesized in yeast extract lysate after in vitro transcription by SP6 polymerase from pGEM4 vectors. Radiolabeled proteins were synthesized for 30 min in the presence of $^{35}$S-labeled Methionine and Cysteine. After translation, the reactions were supplemented with 58 mM 'cold' Methionine-Cysteine mix and 1.5 M Sucrose followed by centrifugation (100,000xg, 60 min, 2 °C) to remove ribosomes and aggregated proteins. The supernatant was diluted with import buffer (250 mM sucrose, 0.25 mg/ml BSA, 80 mM KCl, 5 mM MgCl$_2$, 10 mM MOPS, 2 mM NADH, 2 mM ATP, pH 7.2). Where indicated, the supernatant was supplemented with 20 µM cBag or, as a control, with an equivalent amount of BSA. Isolated mitochondria were diluted in import buffer and supplemented with 4 mM ATP and 2 mM NADH and, where indicated, with 20 µM of C90. Some mitochondria samples were treated with 40 µg/ml of trypsin or proteinase k in SEM buffer (250 mM sucrose, 10 mM MOPS, 1 mM EDTA, pH 7.2) for 30 min followed by 10 min incubation with either soybean trypsin inhibitor (STI) or PMSF, respectively. The import reactions were started by addition of the radiolabeled proteins to the samples containing the isolated organelles and further incubation at 25 °C for the indicated times. The import reactions were stopped by diluting the samples with SEM-K80 buffer (250 mM

sucrose, 80 mM KCl, 10 mM MOPS, 1 mM EDTA, pH 7.2) and re-isolation of mitochondria (13,200xg, 2 min, 2 °C).

The import of the proteins was analyzed by carbonate extraction. To that aim, the re-isolated mitochondria were resuspended in 0.1 M $Na_2CO_3$, incubated on ice for 30 min, and re-isolated by centrifugation (100,000xg, 30 min, 2 °C). The pellets were resuspended in 2 x sample buffer, incubated for 10 min at 95 °C and subjected to SDS-PAGE followed by western blotting and/or autoradiography.

### Fluorescence anisotropy

Peptides corresponding to the transmembrane domains of either Mcr1 or Tom70 were synthesized as described previously (*Jores et al., 2016*). Next, the peptides were coupled to TAMRA and used for fluorescence anisotropy assays. Measurements were performed at 30 °C in a Jasco FP-8500 Fluorospectrometer equipped with polarizers. Excitation and emission wavelength were set to 554 nm and 579 nm, respectively. Samples containing 2 µM TAMRA-labeled peptide were equilibrated for 15 min before 10 µM Ssa1, 30 µM Sis1, 10 µM GST-Tom70, 10 µM BSA or 1 mM ATP in the indicated order or alone were added. For affinity measurements, 2 µM TAMRA-labeled peptides were supplemented with the indicated concentrations of Ssa1 or Sis1 and the difference in anisotropy of bound and free peptide were plotted against the (co)chaperone concentration.

### NanoLC-MS/MS analysis and data processing

Coomassie-stained gel pieces were digested in gel with trypsin, and desalted peptide mixtures (*Rappsilber et al., 2007*) were separated on an Easy-nLC 1200 coupled to a Q Exactive HF mass spectrometer (both Thermo Fisher Scientific) as described elsewhere (*Kliza et al., 2017*) with slight modifications: peptide mixtures were separated using a 57 min segmented gradient of 10-33-50%–90% of HPLC solvent B (80% acetonitrile in 0.1% formic acid) in HPLC solvent A (0.1% formic acid) at a flow rate of 200 nl/min. In each scan cycle, the seven most intense precursor ions were sequentially fragmented using higher energy collisional dissociation (HCD) fragmentation. In all measurements, sequenced precursor masses were excluded from further selection for 30 s. The target values for MS/MS fragmentation were 105 charges, and for the MS scan 3 × 106 charges.

Acquired MS spectra were processed with MaxQuant software package version 1.6.7.0 (*Cox and Mann, 2008*) with integrated Andromeda search engine (*Cox et al., 2011*). Database search was performed against a target-decoy *Saccharomyces cerevisiae* database obtained from Uniprot containing 6,078 protein entries, and 286 commonly observed contaminants. In database search, full trypsin digestion specificity was required and up to two missed cleavages were allowed. Carbamidomethylation of cysteine was set as fixed modification; protein N-terminal acetylation, and oxidation of methionine were set as variable modifications. Initial precursor mass tolerance was set to 4.5 ppm and 20 ppm at the MS/MS level. A false discovery rate of 1% was applied at the peptide and protein level. The mass spectrometry proteomics data have been deposited to the ProteomeXchange Consortium via the PRIDE partner repository with the dataset identifier PXD031610.

## Acknowledgements

We thank E Kracker for excellent technical assistance and T Becker and R Lill for antibodies. This work was supported by the Deutsche Forschungsgemeinschaft (RA 1028/7–2 and RA 1028/10–2 to DR). LD was supported by a long-term fellowship from the Minerva Foundation. We acknowledge support by Open Access Publishing Fund of the University of Tübingen.

## Additional information

### Funding

| Funder | Grant reference number | Author |
| --- | --- | --- |
| Deutsche Forschungsgemeinschaft | RA 1028/7-2 | Doron Rapaport |

| Funder | Grant reference number | Author |
|---|---|---|
| Deutsche Forschungsgemeinschaft | RA 1028/10-2 | Doron Rapaport |
| Minerva Foundation | PhD fellowship | Layla Drwesh |

The funders had no role in study design, data collection and interpretation, or the decision to submit the work for publication.

## Author contributions

Layla Drwesh, Conceptualization, Data curation, Writing - original draft, Writing - review and editing; Benjamin Heim, Data curation, Formal analysis, Writing - review and editing; Max Graf, Linda Kehr, Lea Hansen-Palmus, Data curation, Formal analysis; Mirita Franz-Wachtel, Data curation, Formal analysis, Methodology; Boris Macek, Resources, Software, Supervision, Methodology; Hubert Kalbacher, Data curation, Methodology; Johannes Buchner, Conceptualization, Supervision, Validation; Doron Rapaport, Conceptualization, Resources, Funding acquisition, Writing - review and editing

## Author ORCIDs

Layla Drwesh (iD) http://orcid.org/0000-0002-1868-1487
Johannes Buchner (iD) http://orcid.org/0000-0003-1282-7737
Doron Rapaport (iD) http://orcid.org/0000-0003-3136-1207

## Decision letter and Author response

Decision letter https://doi.org/10.7554/eLife.77706.sa1
Author response https://doi.org/10.7554/eLife.77706.sa2

# Additional files

## Supplementary files

• Supplementary file 1. Proteins that co-purified with in vitro translated Msp1 or Mcr1. (A) List of chaperones that were found in the elution fraction of either Msp1 or Mcr1 but not in the elution of mock pull-down (0). The iBAQ values of the indicated proteins are indicated. (B) A list of chaperones that were enriched in the elution fraction of either Msp1 or Mcr1 as compared to their levels in the elution from mock pull-down assay are indicated. The iBAQ value of each protein in the eluate of the mock pull-down was set to 1 and the relative values in the pull-down assays with wither Msp1 or Mcr1 are indicated.

• MDAR checklist

• Transparent reporting form

## Data availability

The mass spectrometry proteomics data have been deposited to the ProteomeXchange Consortium via the PRIDE partner repository with the dataset identifier PXD031610. All data generated or analyzed during this study are included in the manuscript and supporting file; Source Data files have been provided for all relevant Figures.

The following dataset was generated:

| Author(s) | Year | Dataset title | Dataset URL | Database and Identifier |
|---|---|---|---|---|
| Mirita FW | 2022 | Searching for the interaction partners in yeast cytosol of newly synthesized mitochondrial signal-anchored proteins | http://proteomecentral.proteomexchange.org/cgi/GetDataset?ID=PXD031610 | ProteomeXchange, PXD031610 |

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

# Appendix 1

## Appendix 1—key resources table

| Reagent type (species) or resource | Designation | Source or reference | Identifiers | Additional information |
|---|---|---|---|---|
| Strain, strain background (*Saccharomyces cerevisiae*) | WT | This paper | W303α | |
| Strain, strain background (*Saccharomyces cerevisiae*) | WT | This paper | JSY7452 | |
| Strain, strain background (*Saccharomyces cerevisiae*) | tetO7-Ubi-L-ydj1 | This paper | YMK120α, YDJ1::tetO7-Ubiquitin-Leu-YDJ1 KanMX4 | |
| Strain, strain background (*Saccharomyces cerevisiae*) | tetO7-Ubi-L-sis1 | This paper | YMK120α, SIS1::tetO7-Ubiquitin-Leu-SIS1 His3MX | |
| Strain, strain background (*Saccharomyces cerevisiae*) | tetO7-Ubi-L-ydj1/sis1 | This paper | YMK120α, tetO7-Ubi-Leu-SIS1:HisMX3a; tetO7-Ubi-Leu-YDJ1:KanMX4 | |
| Strain, strain background (*Saccharomyces cerevisiae*) | tom20Δ | This paper | W303α, TOM20::HIS3 | |
| Strain, strain background (*Saccharomyces cerevisiae*) | tom70/71Δ | This paper | JSY7452, TOM70::*TRP1*, TOM71::*HIS3* | |
| Sequence-based reagent | Msp1 Fwd | This paper | PCR primers | GGGGGATCCATGTCTCGCAAATTTGATTTAAAAACGATTACTGATCTTT |
| Sequence-based reagent | Msp1 Rev | This paper | PCR primers | GGGAAGCTTATCAAGAGGTTGAGATGACAACGTACTTG |
| Sequence-based reagent | yk Msp1 Fwd | This paper | PCR primers | GGGGGATCCAAAAAAATGTCTCGCAAATTTGATTTAAAAACGATTACTGATCTTT |
| Sequence-based reagent | Yk Msp1 Rev | This paper | PCR primers | GGGAAGCTTTTAATCAAGAGGTTGAGATGACAAC |
| Sequence-based reagent | yk Msp1-3HA Fwd | This paper | PCR primers | CACACGAGCTCAAAAAAATGTCTCGCAAATTTGATTTAAAAACG |
| Sequence-based reagent | yk Msp1-3HA Rev | This paper | PCR primers | CACACGGATCCCCATCAAGAGGTTGAGATGACAACGTAC |
| Sequence-based reagent | yk Mcr1-3HA Fwd | This paper | PCR primers | GGGGAATTCAAAAAAATGTTTTCCAGATTATCCAGATCTCACTCAAAAGC |
| Sequence-based reagent | yk Mcr1-3HA Rev | This paper | PCR primers | GGGCCCGGGAAATTTGAAAACTTGGTCCTTGGAGTAGCCC |
| Sequence-based reagent | yk Tom20-3HA Fwd | This paper | PCR primers | GGGGGTACCAAAAAAATGTCCCAGTCGAACCCTATCTTAC |
| Sequence-based reagent | yk Tom20-3HA Rev | This paper | PCR primers | GGGGGATCCGGGTCATCGATATCGTTAGCTTCAGC |
| Sequence-based reagent | yk Tom70-3HA Fwd | This paper | PCR primers | GGGGGTACCAAAAAAATGAAGAGCTTCATTACAAGGACAAGAC |
| Sequence-based reagent | yk Tom70-3HA Rev | This paper | PCR primers | GGGGGATCCGGCATTAAACCCTGTTCGCGTAATTTAGC |
| Sequence-based reagent | yk Msp1-TMD-3HA Fwd | This paper | PCR primers | GGGGAATTCAAAAAAATGTCTCGCAAATTTGATTTAAAAACG |
| Sequence-based reagent | yk Msp1-TMD-3HA Rev | This paper | PCR primers | GGGGGATCCCCGTTGAGTAGCCGACTGACCA |

*Appendix 1 Continued on next page*

*Appendix 1 Continued*

| Reagent type (species) or resource | Designation | Source or reference | Identifiers | Additional information |
|---|---|---|---|---|
| Sequence-based reagent | yk Msp1-CD—3HA Fwd | This paper | PCR primers | CACACGAGCTCAAAAAAA TGGATGTTGAATCAGGACCG TTATCAGG |
| Sequence-based reagent | yk Msp1-CD—3HA Rev | This paper | PCR primers | CACACGGATCCCCATCAAG AGGTTGAGATGACAACGTAC TTGTAGC |
| Sequence-based reagent | yk Mcr1-TMD-3HA Fwd | This paper | PCR primers | CACACGAATTCAAAAAAA TGTTTTCCAGATTATCCAG ATCTC |
| Sequence-based reagent | yk Mcr1-TMD-3HA Rev | This paper | PCR primers | CACACCCCGGGGACAAAGG AATGTTGGTTACG GTTT |
| Sequence-based reagent | yk Mcr1-CD-3HA Fwd | This paper | PCR primers | CACACGAATTCAAAAAAAT GCATTCCTTTGTCTTCAATG AATC |
| Sequence-based reagent | yk Mcr1-CD-3HA Rev | This paper | PCR primers | CACACCCCGGGAAATTTGA AAACTTGGTCCTTGGAGTAG |
| Sequence-based reagent | yk Tom20-TMD-3HA Fwd | This paper | PCR primers | GGGGGTACCAAAAAAATGT CCCAGTCGAACCCTATCTTAC |
| Sequence-based reagent | yk Tom20-TMD-3HA Rev | This paper | PCR primers | GGGGGATCCGGGTCAAA GTAGATAGCATAACCGGTG |
| Sequence-based reagent | yk Tom20-CD-3HA Fwd | This paper | PCR primers | GGGGGTACCAAAAAAAT GAGAAATAGCCCGCAATTC AGGAA |
| Sequence-based reagent | yk Tom20-CD-3HA Rev | This paper | PCR primers | GGGGGATCCGGGTCATC GATATCGTTAGCTTCAGC |
| Sequence-based reagent | yk Tom70-TMD-3HA Fwd | This paper | PCR primers | GGGGGTACCAAAAAAA TGAAGAGCTTCATTACAAGGA ACAAGAC |
| Sequence-based reagent | yk Tom70-TMD-3HA Rev | This paper | PCR primers | GGGGGATCCGGCAATTGGT TGTAATAATAGTAGGCACC |
| Sequence-based reagent | yk Tom70-CD-3HA Fwd | This paper | PCR primers | GGGGGTACCAAAAAAATGC AACAACAACAACGAGGAAAA AAGAACAC |
| Sequence-based reagent | yk Tom70-CD-3HA Rev | This paper | PCR primers | GGGGGATCCGGCATTAAACC CTGTTCGCGTAATTTAGC |
| Sequence-based reagent | tetO₇-Ubi-L-Ydj1 Fwd | *Jores et al., 2018* | PCR primers | CATATCTTTTGATAGAACATA ATTAAAAATTATCCAAACTGA ATTCTACACAGTATAGCGACC AGCATTCACATACG |
| Sequence-based reagent | tetO₇-Ubi-L-Ydj1 Rev | *Jores et al., 2018* | PCR primers | GTGGCAGTTACTGGAACACC TAGAATATCGTAAAACTTAG TTTCTTTAACCAAACCACCTC TCAATCTCAAGACCAAG |
| Sequence-based reagent | tetO₇-Ubi-L-Sis1 Fwd | *Jores et al., 2018* | PCR primers | GGATAAGTTGTTTGCATTTTA AGATTTTTTTTTTAATACATT CACATCAACAGTATAGCGAC CAGCATTCACATACG |
| Sequence-based reagent | tetO₇-Ubi-L-Sis1 Rev | *Jores et al., 2018* | PCR primers | TTAGCACTTGGAGATACT CCAAGTAAATCATAAAGTTT TGTCTCCTTGACCAAACCACC TCTCAATCTCAAGACCAAG |
| Recombinant DNA reagent | pGEM4polyA-3HA (plasmid) | *Jores et al., 2018* | | C-terminal 3 x HA-tag |
| Recombinant DNA reagent | pGEM4polyA-yk-DHFR-3HA (plasmid) | *Jores et al., 2018* | | Yeast kozak sequence (AAAAAAATG) DHFR-3 ×HA-tag |
| Recombinant DNA reagent | pGEM4polyA-yk-Porin-3HA (plasmid) | *Jores et al., 2018* | | Yeast kozak sequence (AAAAAAATG) Porin-3 ×HA-tag |
| Recombinant DNA reagent | pGEM4polyA-yk-Fis1-3HA (plasmid) | This paper | | Yeast kozak sequence (AAAAAAATG) Fis1−3×HA-tag |
| Recombinant DNA reagent | pGEM4polyA-yk-Msp1-3HA (plasmid) | This paper | | Yeast kozak sequence (AAAAAAATG) Msp1−3×HA-tag |

*Appendix 1 Continued on next page*

*Appendix 1 Continued*

| Reagent type (species) or resource | Designation | Source or reference | Identifiers | Additional information |
|---|---|---|---|---|
| Recombinant DNA reagent | pGEM4polyA-yk-Mcr1-3HA (plasmid) | This paper | | Yeast kozak sequence (AAAAAAATG) Mcr1–3×HA-tag |
| Recombinant DNA reagent | pGEM4polyA-yk-Tom20-3HA (plasmid) | This paper | | Yeast kozak sequence (AAAAAAATG) Tom20–3×HA-tag |
| Recombinant DNA reagent | pGEM4polyA-yk-Tom70-3HA (plasmid) | This paper | | Yeast kozak sequence (AAAAAAATG) Tom70–3×HA-tag |
| Recombinant DNA reagent | pGEM4polyA-yk-Msp1(33-363)–3 HA (plasmid) | This paper | | Yeast kozak sequence (AAAAAAATG) Msp1(1-363)–3×HA-tag |
| Recombinant DNA reagent | pGEM4polyA-yk-Msp1(1-32)–3 HA (plasmid) | This paper | | Yeast kozak sequence (AAAAAAATG) Msp1(1-32)–3×HA-tag |
| Recombinant DNA reagent | pGEM4polyA-yk-Mcr1(35-302)–3 HA (plasmid) | This paper | | Yeast kozak sequence (AAAAAAATG) Mcr1(35-302)–3×HA-tag |
| Recombinant DNA reagent | pGEM4polyA-yk-Mcr1(1-39)–3 HA (plasmid) | This paper | | Yeast kozak sequence (AAAAAAATG) Mcr1(1-39)–3×HA-tag |
| Recombinant DNA reagent | pGEM4polyA-yk-Tom20(33-183)–3 HA (plasmid) | This paper | | Yeast kozak sequence (AAAAAAATG) Tom20(33-183)–3×HA-tag |
| Recombinant DNA reagent | pGEM4polyA-yk-Tom20(1-30)–3 HA (plasmid) | This paper | | Yeast kozak sequence (AAAAAAATG) Tom20(1-30)–3×HA-tag |
| Recombinant DNA reagent | pGEM4polyA-yk-Tom70(33-617)–3 HA (plasmid) | This paper | | Yeast kozak sequence (AAAAAAATG) Tom70(33-617)–3×HA-tag |
| Recombinant DNA reagent | pGEM4polyA-yk-Tom70(1-32)–3 HA (plasmid) | This paper | | Yeast kozak sequence (AAAAAAATG) Tom70(1-32)–3×HA-tag |
| Recombinant DNA reagent | pMK632His (plasmid) | *Jores et al., 2018* | | HIS3MX cassette tetO7-CYC1 promoter-Ubiquitin-Leucin-HA-tag |
| Recombinant DNA reagent | pMK632Kan (plasmid) | *Jores et al., 2018* | | KanMX cassette tetO7-CYC1 promoter-Ubiquitin-Leucin-HA-tag |
| Recombinant DNA reagent | pGEX4T1-GST (plasmid) | This paper | | GST |
| Recombinant DNA reagent | pGEX4T1-GST-Tom20(35-183) (plasmid) | This paper | | Tom20(35-183) |
| Recombinant DNA reagent | pGEX4T1-GST-Tom70 (46-617) (plasmid) | This paper | | Tom70(46-617) |
| Recombinant DNA reagent | pPROEX-HTa-cBag (plasmid) | *Young et al., 2003* | | His6-tag-TEV-human Bag-1M(151-263) |
| Recombinant DNA reagent | pPROEX-HTa-(C90) (plasmid) | *Young et al., 2003* | | His6-tag-TEV-human Hsp90a(566-732) |
| Antibody | Anti-Ssa1/2 (rabbit polyclonal) | *Jores et al., 2018* | | 1:20,000 |
| Antibody | Anti-Ydj1 (rabbit polyclonal) | *Jores et al., 2018* | | 1:10,000 |
| Antibody | Anti-Sis1 (rabbit polyclonal) | *Jores et al., 2018* | | 1:20,000 |
| Antibody | Anti-Hsp26 (rabbit polyclonal) | *Jores et al., 2018* | | 1:4000 |
| Antibody | Anti-Hsp104 (rabbit polyclonal) | *Jores et al., 2018* | | 1:25,000 |
| Antibody | Anti-Hsp42 (rabbit polyclonal) | *Jores et al., 2018* | | 1:4000 |
| Antibody | Anti-Hsp82 (rabbit polyclonal) | *Jores et al., 2018* | | 1:20,000 |
| Antibody | Anti-Hch1 (rabbit polyclonal) | *Jores et al., 2018* | | 1:4000 |
| Antibody | Anti-Bmh1 (rabbit polyclonal) | This paper | | 1:1000 |

*Appendix 1 Continued on next page*

*Appendix 1 Continued*

| Reagent type (species) or resource | Designation | Source or reference | Identifiers | Additional information |
|---|---|---|---|---|
| Antibody | Anti-Djp1 (rabbit polyclonal) | Lab of Ineke Braakman | | 1:2000 |
| Antibody | Anti-Sti1 (rabbit polyclonal) | This paper | | 1:10,000 |
| Antibody | Anti-Aha1 (rabbit polyclonal) | This paper | | 1:2000 |
| Antibody | Anti-Msp1 (rabbit polyclonal) | Lab of Toshiya Endo | | 1:2000 |
| Antibody | Anti-Mcr1 (rabbit polyclonal) | This paper | | 1:2000 |
| Antibody | Anti-Fis1 (rabbit polyclonal) | This paper | | 1:1000 |
| Antibody | Anti-Tom20 (rabbit polyclonal) | This paper | | 1:4000 |
| Antibody | Anti-Tom70 (rabbit polyclonal) | This paper | | 1:5000 |
| Antibody | Anti-Porin (rabbit polyclonal) | This paper | | 1:6000 |
| Antibody | Anti-Pic2 (rabbit polyclonal) | This paper | | 1:2000 |
| Antibody | Anti-HA (rat polyclonal) | Roche | #11867423001 | 1:1000 |
| Antibody | Anti-Cor1 (rabbit polyclonal) | This paper | | 1:2000 |
| Antibody | Anti-Mim1 (rabbit polyclonal) | This paper | | 1:100 |
| Antibody | Anti-Cox2 (rabbit polyclonal) | This paper | | 1:2000 |
| Antibody | Anti-Oxa1 (rabbit polyclonal) | This paper | | 1:2000 |
| Antibody | Anti-Erv1 (rabbit polyclonal) | Lab of Johannes Herrmann | | 1:1000 |
| Antibody | Anti-Aco1 (rabbit polyclonal) | This paper | | 1:2000 |
| Antibody | Anti-Tom22 (rabbit polyclonal) | This paper | | 1:2000 |
| Antibody | Anti-Om14 (rabbit polyclonal) | Lab of Thomas Becker | | 1:2000 |
| Antibody | Goat Anti-Rabbit IgG HRP conjugate | Bio-Rad | #1721019 | 1:10,000 |
| Antibody | Goat Anti-Rat IgG HRP conjugate | Abcam | #ab6845 | 1:2000 |

