## [Editor Report]

The authors dissect and reconstitute the cytosolic steps of biogenesis of mitochondrial signal-anchored membrane proteins, focusing on post-translational precursor recognition by cytosolic chaperones and their subsequent transfer to mitochondrial import receptors. These are crucial events in order to assist proper protein biogenesis while preventing aggregation and its downstream consequences. The study is an important contribution to the understanding of cytosolic events in the biogenesis of mitochondrial proteins and will be of relevance for researchers in the fields of chaperone and mitochondrial biology as well as for cell biologists studying the biogenesis of membrane proteins.

---

## [Decision Letter]

**Decision letter after peer review:**

Thank you for submitting your article "A network of cytosolic (co)chaperones promotes the biogenesis of mitochondrial signal-anchored outer membrane proteins" for consideration by *eLife*. Your article has been reviewed by 3 peer reviewers, and the evaluation has been overseen by Klaus Pfanner as the Reviewing Editor and Vivek Malhotra as the Senior Editor. The following individuals involved in the review of your submission have agreed to reveal their identity: Thomas Becker (Reviewer #1); Nils Wiedemann (Reviewer #2); Kostas Tokatlidis (Reviewer #3).

Essential revisions:

1) The authors preincubated mitochondria with C90 to block Tom70 function. To directly show the role of Tom70, I suggest including tom70delta mitochondria in the import experiments in figure 9.

2) The interesting in vivo experiments assessing the phenotype of Sis1 and/or Ydj1 downregulation (Figure 3) are among the crucial experiments of the manuscript, revealing a reduction of Tom20 and Tom70 receptor levels upon Sis1 downregulation. Since α-helical outer membrane proteins have differing dependencies on MIM and TOM core complex, the authors should test if the import machineries required for efficient biogenesis of α-helical outer membrane proteins are not similarly affected compared to Tom20 and Tom70. Otherwise, one could not exclude that the defects observed for Tom20 and Tom70 are only secondary for example by diminished Mim1 levels.

3) The authors should label the Mcr1 MOM and MIM isoforms (Figure 3). It looks like that in contrast to the Mcr1 MOM isoform, which does not seem to be affected by Sis1/Yid1 depletion, the Mcr1 MIM isoform seems to be affected. Thus, it would be conceivable that the cytosolic Hsp40s Sis1/Yid1 are not only required for the biogenesis of α-helical outer membrane proteins, but also for the biogenesis of α-helical (stop transfer) inner membrane proteins. To analyze if Sis1/Ydj1 are generally required for the biogenesis of α-helical membrane proteins, the authors should blot for further α-helical membrane proteins of the outer and inner membranes. To evaluate the specificity of the Sis1/Ydj1 system, the authors should also check the integrity and/or activity of the oxidative phosphorylation complexes.

4) End of p 13, they describe trypsin shaving of the surface of mitochondria. Strictly speaking, this will not get rid of all OM receptors in mitochondria (some are trypsin resistant), so maybe a repetition of this experiment adding PK (proteinase K) as well, might be more conclusive and remove any potential redundant effects of trypsin-resistant receptors.

5) As this is a very complex topic, particularly for non-experts, it might be worthwhile having a model cartoon at the end to bring all the info together in a summary model.

*Reviewer #2 (Recommendations for the authors):*

1) The interesting in vivo experiments assessing the phenotype of Sis1 and/or Ydj1 downregulation (Figure 3) are among the most crucial experiments of the manuscript, revealing a reduction of Tom20 and Tom70 receptor levels upon Sis1 downregulation. Since α-helical outer membrane proteins have differing dependencies on MIM and TOM core complex the authors should test if the import machineries required for efficient biogenesis of α-helical outer membrane proteins are not similarly affected compared to Tom20 and Tom70. Otherwise, one could not exclude, that the defects observed for Tom20 and Tom70 are only secondary caused for example by diminished Mim1 levels.

2) In addition, the authors should label the Mcr1 MOM and MIM isoforms (Figure 3). It looks like that in contrast to the Mcr1 MOM isoform, which does not seem to be affected by Sis1/Yid1 depletion, the Mcr1 MIM isoform seems to be affected. Thus, it would be conceivable, that the cytosolic Hsp40's Sis1/Yid1 are not only required for the biogenesis of α-helical outer membrane proteins, but also for the biogenesis of α-helical (stop transfer) inner membrane proteins. To analyze if Sis1/Ydj1 are generally required for the biogenesis of α-helical membrane proteins, the authors should blot for many further α-helical membrane proteins of the outer and inner membranes. To evaluate the specificity of the Sis1/Ydj1 system, the authors should also check the integrity and/or activity of the oxidative phosphorylation complexes.

3) To stress the general relevance of the cytosolic Sis1/Ydj1 system for the biogenesis of α-helical outer membrane proteins, the authors should test further substrates employing the in vitro import assay into isolated mitochondria with precursor proteins synthesized from extracts depleted of Ydj1 and Sis1 (YS down, see Figure 4).

4) To stress the general relevance of the cytosolic Hsp70 system for the biogenesis of α-helical outer membrane proteins, the authors should test further substrates employing the in vitro import assay into isolated mitochondria with precursor proteins in the absence and presence of CBag (see Figure 7A and B).

5) How do the authors explain that the biogenesis of Mcr1 is not inhibited by Trypsin pretreatment of isolated mitochondria, but by importing in the presence of C90 into tom20delta mitochondria (Figure 9).

---

## [Author Response]

Essential revisions:1) The authors preincubated mitochondria with C90 to block Tom70 function. To directly show the role of Tom70, I suggest including tom70delta mitochondria in the import experiments in figure 9.

To address this point, we performed experiments with mitochondria isolated from cells codeleted for TOM70 and its paralogue TOM71. The results of these import assays, which are presented in revised Figure S2A (for Msp1) and Figure S2B (for Mcr1), show that the import of both signal anchored proteins is hardly affected. We suggest that the apparent contradiction to the reduction in the import of these substrates observed upon addition of C90 is due to potential compensation processes in the deletion strain whereas the addition of C90 does allow adaptation time. This point is further discussed in the text.

2) The interesting in vivo experiments assessing the phenotype of Sis1 and/or Ydj1 downregulation (Figure 3) are among the crucial experiments of the manuscript, revealing a reduction of Tom20 and Tom70 receptor levels upon Sis1 downregulation. Since α-helical outer membrane proteins have differing dependencies on MIM and TOM core complex, the authors should test if the import machineries required for efficient biogenesis of α-helical outer membrane proteins are not similarly affected compared to Tom20 and Tom70. Otherwise, one could not exclude that the defects observed for Tom20 and Tom70 are only secondary for example by diminished Mim1 levels.

To address this justified comment, we monitored the levels of various proteins in the strain depleted for both Ydj1 and Sis1. The outcome of these immunodecorations, as presented in revised Figure S1A, demonstrate that, as we reported before (Jores et al., 2018), the double depletion of both co-chaperones results in major reduction in the levels of the β-barrel proteins Tom40 and porin. Interestingly, the levels of the inner membrane protein Oxa1 are also compromised whereas other inner membrane proteins like Cor1 or Cox2 appear unaffected. Importantly, the amounts of Mim1 as analyzed by SDS-PAGE or BN-PAGE are not influenced by the reduced levels of both co-chaperones (Figure S1A and B) suggesting that the observed reduction in the levels of Tom20 and Tom70 cannot be explained by compromised activity of the MIM insertase. These new results are discussed in the revised text.

3) The authors should label the Mcr1 MOM and MIM isoforms (Figure 3). It looks like that in contrast to the Mcr1 MOM isoform, which does not seem to be affected by Sis1/Yid1 depletion, the Mcr1 MIM isoform seems to be affected. Thus, it would be conceivable that the cytosolic Hsp40s Sis1/Yid1 are not only required for the biogenesis of α-helical outer membrane proteins, but also for the biogenesis of α-helical (stop transfer) inner membrane proteins. To analyze if Sis1/Ydj1 are generally required for the biogenesis of α-helical membrane proteins, the authors should blot for further α-helical membrane proteins of the outer and inner membranes. To evaluate the specificity of the Sis1/Ydj1 system, the authors should also check the integrity and/or activity of the oxidative phosphorylation complexes.

As requested, we labeled in Figure 3 the two isoforms of Mcr1.

To address these valid points, we immunodecorated against several helical proteins of the outer and inner mitochondrial membranes. It seems that various proteins depend to a variable extent on both co-chaperones. Whereas the OM proteins Om14, Tom22, Tom20, and Tom70 are affected upon the dual depletion of Ydj1 and Sis1, the levels of other OM proteins like Fis1, Msp1, or Mcr1(OM form) were not altered. Similar variability was observed also for inner membrane or IMS proteins. Cor1, Cox2, and Pic2 were not affected while Oxa1 and Mcr1(IMS form) were detected in reduced amounts (revised Figure 3D). Analyzing the respiratory complexes III and IV by BN-PAGE did not reveal major differences upon depletion of the co-chaperones (Figure S1B). Hence, it seems that the biogenesis of some (like Tom20, Tom70, Oxa1 or Mcr1_IMS_) but not all membranal mitochondrial proteins are supported by Ydj1 and Sis1. These new observations are presented and discussed in the revised text.

4) End of p 13, they describe trypsin shaving of the surface of mitochondria. Strictly speaking, this will not get rid of all OM receptors in mitochondria (some are trypsin resistant), so maybe a repetition of this experiment adding PK (proteinase K) as well, might be more conclusive and remove any potential redundant effects of trypsin-resistant receptors.

The reviewer raised a valid point. To address this issue, we performed many experiments where we compare the import behavior of mitochondria pre-treated before the import reactions with either PK or trypsin. Surprisingly, we observed a stronger effect upon treatment with trypsin although in both cases the import receptors Tom20 and Tom70 were removed (revised Figure 9A and B). The effect of trypsin in the new assays was much higher than in the original submission probably due to the usage in the new experiments of a digestion buffer lacking BSA. We suspect that in the original experiments a large portion of the trypsin proteolytic capacity was toward BSA instead of efficient digestion of the mitochondrial proteins. Although we cannot explain completely the difference between trypsin and PK, we included in the revised text some suggestions.

5) As this is a very complex topic, particularly for non-experts, it might be worthwhile having a model cartoon at the end to bring all the info together in a summary model.

This is a very good suggestion and accordingly we included in the revised version (as revised Figure 10) a scheme that depicts our working model.

Reviewer #2 (Recommendations for the authors):1) The interesting in vivo experiments assessing the phenotype of Sis1 and/or Ydj1 downregulation (Figure 3) are among the most crucial experiments of the manuscript, revealing a reduction of Tom20 and Tom70 receptor levels upon Sis1 downregulation. Since α-helical outer membrane proteins have differing dependencies on MIM and TOM core complex the authors should test if the import machineries required for efficient biogenesis of α-helical outer membrane proteins are not similarly affected compared to Tom20 and Tom70. Otherwise, one could not exclude, that the defects observed for Tom20 and Tom70 are only secondary caused for example by diminished Mim1 levels.

Please refer to point #5 in the essential issues above.

2) In addition, the authors should label the Mcr1 MOM and MIM isoforms (Figure 3). It looks like that in contrast to the Mcr1 MOM isoform, which does not seem to be affected by Sis1/Yid1 depletion, the Mcr1 MIM isoform seems to be affected. Thus, it would be conceivable, that the cytosolic Hsp40's Sis1/Yid1 are not only required for the biogenesis of α-helical outer membrane proteins, but also for the biogenesis of α-helical (stop transfer) inner membrane proteins. To analyze if Sis1/Ydj1 are generally required for the biogenesis of α-helical membrane proteins, the authors should blot for many further α-helical membrane proteins of the outer and inner membranes. To evaluate the specificity of the Sis1/Ydj1 system, the authors should also check the integrity and/or activity of the oxidative phosphorylation complexes.

Please refer to point #5 in the essential issues above.

3) To stress the general relevance of the cytosolic Sis1/Ydj1 system for the biogenesis of α-helical outer membrane proteins, the authors should test further substrates employing the in vitro import assay into isolated mitochondria with precursor proteins synthesized from extracts depleted of Ydj1 and Sis1 (YS down, see Figure 4).

Since we observed a reduction of the steady-state levels of Tom20 and Tom70 in mitochondria isolated from these double depleted cells, we focused in our in vitro import assays on these substrate proteins. The idea to perform additional import assays is good but is beyond the scope of the current contribution.

4) To stress the general relevance of the cytosolic Hsp70 system for the biogenesis of α-helical outer membrane proteins, the authors should test further substrates employing the in vitro import assay into isolated mitochondria with precursor proteins in the absence and presence of CBag (see Figure 7A and B).

Additional import assays could contribute to the generality of our findings but are beyond the scope of the current contribution.

5) How do the authors explain that the biogenesis of Mcr1 is not inhibited by Trypsin pretreatment of isolated mitochondria, but by importing in the presence of C90 into tom20delta mitochondria (Figure 9).

The trypsin treatment in the original experiments was done with a buffer that contained too much BSA. Hence, the trypsin was occupied with BSA and the effect was less than expected. In the revised version we include new experiments that were done with a buffer lacking BSA. In these experiments (revised Figure 9) the effect of trypsin is clear and matches the effect of C90.